# Dynamic coordination engineering of 2D PhenPtCl$_2$ nanosheets for superior hydrogen evolution

Gonglei Shao [1,9] ✉, Changfei Jing [2,3,9], Zhinan Ma [4], Yuanyuan Li [5], Weiqi Dang [6], Dong Guo [1], Manman Wu [1], Song Liu [7], Xu Zhang [1], Kun He [2], Yifei Yuan [2], Jun Luo [8], Sheng Dai [3] ✉, Jie Xu [2] ✉ & Zhen Zhou [1] ✉

Exploring the dynamic structural evolution of electrocatalysts during reactions represents a fundamental objective in the realm of electrocatalytic mechanism research. In pursuit of this objective, we synthesized PhenPtCl$_2$ nanosheets, revealing a N$_2$-Pt-Cl$_2$ coordination structure through various characterization techniques. Remarkably, the electrocatalytic performance of these PhenPtCl$_2$ nanosheets for hydrogen evolution reaction (HER) surpasses that of the commercial Pt/C catalyst across the entire pH range. Furthermore, our discovery of the dynamic coordination changes occurring in the N$_2$-Pt-Cl$_2$ active sites during the electrocatalytic process, as clarified through in situ Raman and X-ray photoelectron spectroscopy, is particularly noteworthy. These changes transition from Phen-Pt-Cl$_2$ to Phen-Pt-Cl and ultimately to Phen-Pt. The Phen-Pt intermediate plays a pivotal role in the electrocatalytic HER, dynamically coordinating with Cl$^-$ ions in the electrolyte. Additionally, the unsaturated, two-coordinated Pt within Phen-Pt provides additional space and electrons to enhance both H$^+$ adsorption and H$_2$ evolution. This research illuminates the intricate dynamic coordination evolution and structural adaptability of PhenPtCl$_2$ nanosheets, firmly establishing them as a promising candidate for efficient and tunable electrocatalysts.

Electrocatalysis plays a pivotal role in clean energy research, offering promising solutions for sustainable energy conversion and storage. The hydrogen evolution reaction (HER) stands as a crucial electrochemical process with profound implications for clean hydrogen production and fuel cell technology[1–3]. Platinum (Pt)-based catalysts have showcased exceptional performance in the HER, but their high cost and limited availability have impeded their widespread adoption[4–6]. Consequently, there is a mounting interest

[1]Interdisciplinary Research Center for Sustainable Energy Science and Engineering (IRC4SE2), School of Chemical Engineering, Zhengzhou University, Zhengzhou 450001, PR China. [2]College of Chemistry and Materials Engineering, Wenzhou University, Wenzhou 325035, PR China. [3]Feringa Nobel Prize Scientist Joint Research Centre, School of Chemistry and Molecular Engineering, East China University of Science and Technology, Shanghai 200237, PR China. [4]School of Chemistry and Chemical Engineering, North University of China, Taiyuan 030051 Shanxi, PR China. [5]School of Sciences, Henan University of Technology, Zhengzhou 450001, PR China. [6]National Laboratory of Solid State Microstructures, School of Physics, Collaborative Innovation Center of Advanced Microstructures, Nanjing University, Nanjing 210093, PR China. [7]Institute of Chemical Biology and Nanomedicine (ICBN), State Key Laboratory of Chemo/Biosensing and Chemometrics, College of Chemistry and Chemical Engineering, Hunan University, Changsha 410082, PR China. [8]ShenSi Lab, Shenzhen Institute for Advanced Study, University of Electronic Science and Technology of China, Longhua District, Shenzhen 518110, PR China. [9]These authors contributed equally: Gonglei Shao, Changfei Jing. ✉e-mail: shaogonglei@zzu.edu.cn; shengdai@ecust.edu.cn; jiexu@wzu.edu.cn; zhouzhen@nankai.edu.cn

in investigating alternative, cost-effective catalysts capable of matching or even outperforming the performance of commercial Pt/C catalysts.

In recent years, two-dimensional (2D) materials have garnered attention as potential alternatives to Pt-based catalysts due to their unique properties, high surface area, and tunable electronic/atomic structures[7–9]. Despite the potential of 2D materials, there are still challenges in understanding the intricate relationship between their structures and electrocatalytic performance[10–12]. Particularly, the dynamic structural evolution of 2D materials during the HER process and the influence on the catalytic efficiency and stability remain unclear. In the realm of electrocatalysis, understanding the dynamic and accurate structure-property relationship stands as the ultimate goal to the research of catalytic mechanisms[13,14]. Exploring how the atom arrangement influences the electrochemical activity and efficiency of the catalyst is of paramount importance[15].

Among these alternatives, 2D PhenPtCl$_2$ nanosheets, a molecular crystal condensed from PhenPtCl$_2$ molecules, feature an asymmetric coordinated Pt with two Cl$^-$ ions and a phenanthroline (Phen) bidentate ligand. Notably, the catalytic potential of Pt as the active center can be finely tuned through ligands. Furthermore, the presence of Cl$^-$ provides an unparalleled level of control over the coordination structure of 2D PhenPtCl$_2$ nanosheets, allowing for a dynamic coordination number of Pt within the range of 2 to 4. This flexibility in coordination structure, combined with the ease of dynamic coordination, enables the assessment of catalytic performances at various stages with advanced characterization techniques. This, in turn, facilitates the analysis of the effects of coordination structure, valence state, and surface charge distribution of active metal sites on catalytic performance.

Through an exploration of the dynamic coordination changes during the electrocatalytic process, researchers can unveil the key intermediates and coordination configurations that govern catalytic prowess[16–19]. Ultimately, this pursuit of understanding structure-performance relationships propels the development of high-performance and versatile electrocatalysts, ushering in a new era of sustainable and efficient energy conversion technologies. As researchers continue to delve deeper into this dynamic interplay, catalytic mechanism exploration takes us one step closer to unlocking the full potential of electrocatalysis in addressing global energy challenges.

In this study, we chose 2D PhenPtCl$_2$ nanosheets as a prime example for investigating their suitability in the HER across the entire pH range. These nanosheets exhibit a unique crystal structure characterized by the N$_2$-Pt-Cl$_2$ coordination, which positions them as promising candidates for electrocatalysis. We employed advanced low-dose integrated differential phase contrast-scanning transmission electron microscopy (iDPC-STEM) to unveil the distinctive crystal structure of 2D PhenPtCl$_2$ nanosheets. Throughout the HER electrocatalysis, we harnessed sophisticated characterization techniques, including in situ Raman spectroscopy, in situ X-ray photoelectron spectroscopy (XPS), and X-ray absorption spectroscopy (XAS), to dissect the dynamic coordination changes occurring in the Pt active sites. Our findings revealed a dynamic coordination transformation progressing from Phen-Pt-Cl$_2$ to Phen-Pt-Cl and finally to Phen-Pt. The dynamically unsaturated coordination structure of Pt in the intermediate state, Phen-Pt, serves as the linchpin for the exceptional electrocatalytic HER performance of 2D PhenPtCl$_2$ nanosheets, with the two-coordinated Pt offering additional space and electrons to enhance H$^+$ adsorption and hydrogen evolution. This research is of great significance, shedding light on how the crystal structure of these nanosheets influences their electrocatalytic activity and stability. Ultimately, the insights gained from this study contribute to the advancement in the field of electrocatalysis and the development of sustainable energy technologies.

## Results

### Synthesis and characterization of 2D PhenPtCl$_2$ nanosheets

The 2D PhenPtCl$_2$ molecular crystal was synthesized from 1,10-Phen and chloroplatinic acid (H$_2$PtCl$_6$) with the assistance of ultrasonic waves in solution (Fig. 1a). The resulting 2D PhenPtCl$_2$ crystal presents a four-coordination atomic structure at the Pt site, comprising a Phen molecule and two Cl$^-$ ligands. To confirm its 2D morphology, ultrathin flake-like samples were captured in optical images (as depicted in the inset of Fig. 1b). Additional optical images (Supplementary Fig. 1) and low-magnification STEM images (Supplementary Fig. 2) and scanning electron microscopy (SEM) images (Supplementary Fig. 3) further validate the nanosheet's regular quadrilateral structure, characteristic of the 2D PhenPtCl$_2$ crystal. The thickness of these nanosheets was determined to be 8.3 nm through atomic force microscopy (AFM), as depicted in Fig. 1b.

Obviously, the Raman spectroscopy of 2D PhenPtCl$_2$ crystal manifests three new vibration peaks at 339.7 cm$^{-1}$, 316.2 cm$^{-1}$ and 156.2 cm$^{-1}$, corresponding to the stretching vibration peaks of Cl-Pt, N-Pt and C-N bonds, respectively[20]. In comparison to the Raman peaks of the Phen precursor, the peak at 247.1 cm$^{-1}$, associated with the C=N double bond, disappears in 2D PhenPtCl$_2$ crystal, while the peak at 411.3 cm$^{-1}$ corresponding to the C−C bond remains[21]. These transformations indicate the conversion of the C=N double bond into a C-N bond, forming a new coordination bond with Pt (Fig. 1c). Infrared (IR) spectroscopy also depicts the changes in the stretching vibrations of the organic skeleton (Fig. 1d). After synthesis, the absorption peaks corresponding to the Cl (536.1, 714.8, and 776.3 cm$^{-1}$), C−N (1188.1 cm$^{-1}$) and −N−Pt (1536.9 cm$^{-1}$) can be clearly observed, while the absorption peaks of the C=N functional group (1089.41 and 1642.7 cm$^{-1}$) disappear[22]. These observations suggest the electron redistribution around the Pt center, i.e., the opening of the C=N double bond, and the bonding formation of Pt with N and Cl. Furthermore, the appearance of O-H functional groups (3384.1 cm$^{-1}$) is attributed to the presence of molecular water in the Phen crystal. The appearance of N-H (3209.6 cm$^{-1}$) is introduced by the reaction of H$^+$ from chloroplatinic acid solution with tertiary amines, resulting in the formation of quaternary ammonium salts[23]. Detailed solid-state nuclear magnetic resonance (NMR) analyses of C and H spectra show that only a slight shift occurs relative to Phen organic crystals, indicating the coordination between Pt and Phen (Supplementary Fig. 4)[24,25]. For the analysis of the single crystal structure, the 2D PhenPtCl$_2$ crystal possesses the lattice constants, $a = 9.53$ Å, $b = 17.12$ Å, and $c = 7.26$ Å, along with $\alpha = 90°$, $\beta = 109°$ and $\gamma = 90°$. Supplementary Fig. 5 illustrates the atomic model of PhenPtCl$_2$ crystal observed from a, b and c crystal axis directions. The energy dispersive spectroscopy (EDS) elemental maps also fully indicate the existence and uniform distribution of the four elements (Fig. 1e).

The crystal structure of 2D PhenPtCl$_2$ at room temperature (25 °C) was further characterized by aberration-corrected STEM (AC-STEM). First, the atomic resolution high-angle annular dark-field-STEM (HAADF-STEM) image does not show the fine atomic structure of PhenPtCl$_2$ (Supplementary Fig. 6a), since the organic molecular structure of the PhenPtCl$_2$ crystal is sensitive to electron beam irradiation. However, the corresponding selected area electron diffraction (SAED) patterns of 2D PhenPtCl$_2$ also indicate its single crystal structure (Supplementary Fig. 6c). To reveal the atomic configuration of 2D PhenPtCl$_2$ nanosheets, advanced iDPC-STEM technology was performed. As depicted in Fig. 1f, g, the iDPC-STEM images exhibit a clear and well-defined periodic arrangement of the crystal structure (more iDPC-STEM images in Supplementary Fig. 7). As shown in Fig. 1g, the unit length of the PhenPtCl$_2$ structure is 0.56 nm, consistent with its molecular model. The above results clearly show the integrity and periodicity of the 2D PhenPtCl$_2$ crystal.

To further reveal the 2D characteristics of the PhenPtCl$_2$ crystal, the relationship between the molar ratio of Pt to Phen and the resulting thickness and morphology was investigated. When the ratio of Pt: Phen

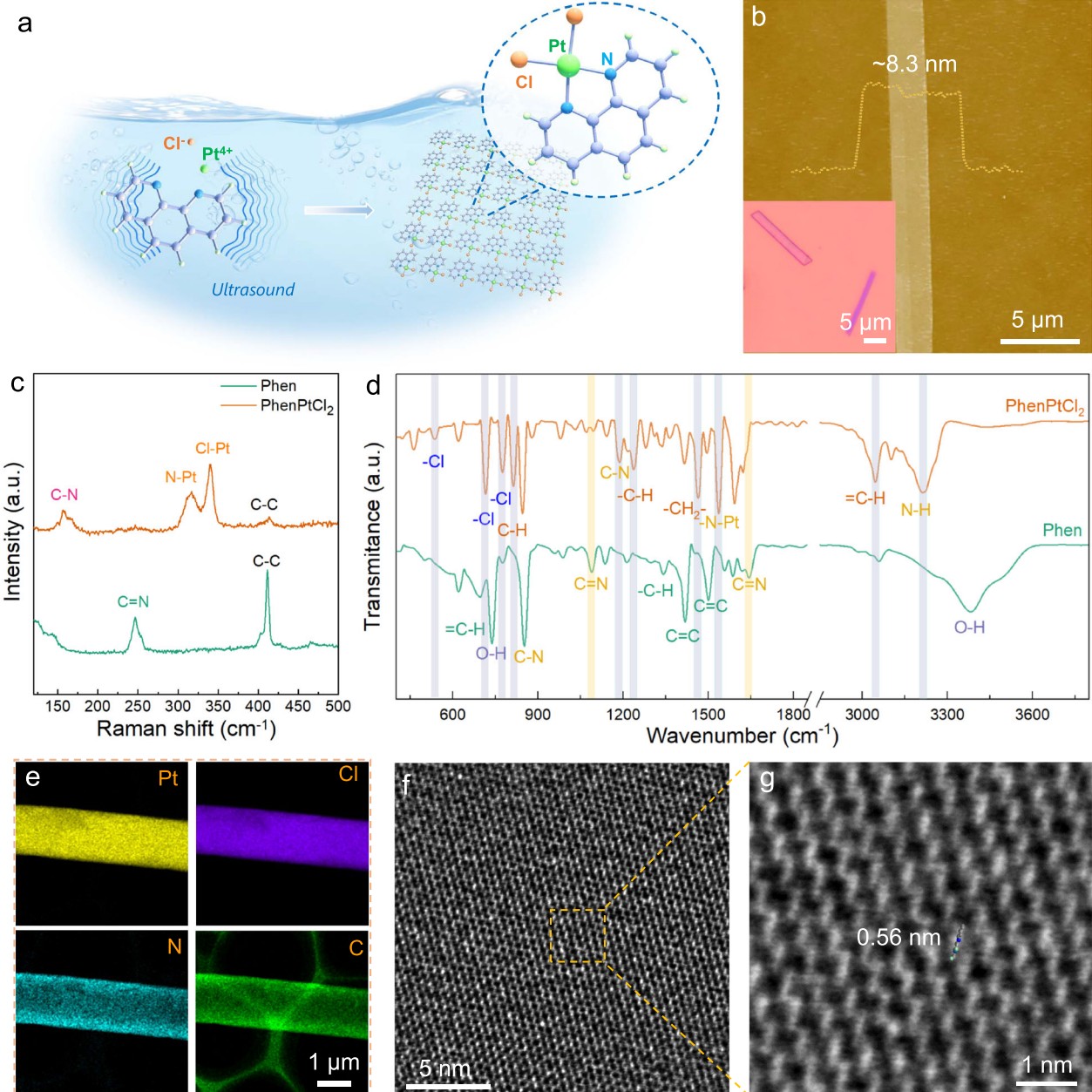

**Fig. 1 | Structural analysis of the 2D PhenPtCl₂ crystal. a** Schematic diagram for the synthesis of 2D PhenPtCl₂ nanosheets. **b** AFM image of 2D PhenPtCl₂ nanosheets at 25 °C. Inset: optical image of 2D PhenPtCl₂ nanosheets. **c** Raman and (**d**) IR spectroscopy of 2D PhenPtCl₂ crystal at 25 °C compared with Phen crystal. **e** EDS mapping of 2D PhenPtCl₂ nanosheets at 25 °C. **f** iDPC image and (**g**) enlarged iDPC image (false-color) of 2D PhenPtCl₂ at 25 °C.

decreases from 10: 1 to 1: 4, the thickness of 2D PhenPtCl₂ nanosheets gradually increases up to 177.3 nm. While the length/width ratio decreases from approximately 18 to around 1, and gradually increases by about 5 (Supplementary Figs. 8–10). These phenomena indicate that 2D Phen molecules play a crucial role in controlling the thickness and morphology of 2D PhenPtCl₂ nanosheets, and the inherent 2D properties of Phen molecule drive the growth along the 2D planes, in accordance with the 2D layer growth theory of crystals[26,27]. Meanwhile, thermogravimetric-mass spectrometry (TG-MS) was employed to investigate the weight loss behavior of the PhenPtCl₂ crystal under elevated temperature (Fig. 2a). Compared with the complete weight loss of the Phen crystal at approximately 260 °C (the weight loss around 100 °C is attributed to the removal of crystal water), the crystal weight loss of PhenPtCl₂ crystal shows two distinct stages. The first weight loss occurs from 218.2 to 277.7 °C, with a corresponding loss

rate of 32.06 %. This weight loss is speculated to be caused by the removal of Phen ligands. The second weight loss happens between 351.1 and 401.4 °C, with a loss rate of 9.95 %, indicating the loss of Cl⁻. Meanwhile, the mass spectrum fragments (m/z) that can be detected mainly include 17 and 18 ($H_2O$), 12 and 44 ($CO_2$), and other fragments, while the detection of 35 (Cl) is weak, but still exists[28,29]. Notably, the mass spectrum fragment signal corresponding to the thermal decomposition product Phen, which is expected to occur between 200 and 400 °C, was not detected. This absence may be attributed to the relatively high boiling point of the escaped component, which could condense in the transmission tube and consequently not reach the detector of the mass spectrum. Therefore, the structural changes at different annealing temperatures inevitably lead to the difference in HER performance, which is important for later optimizing HER performance of the 2D PhenPtCl₂ crystal.

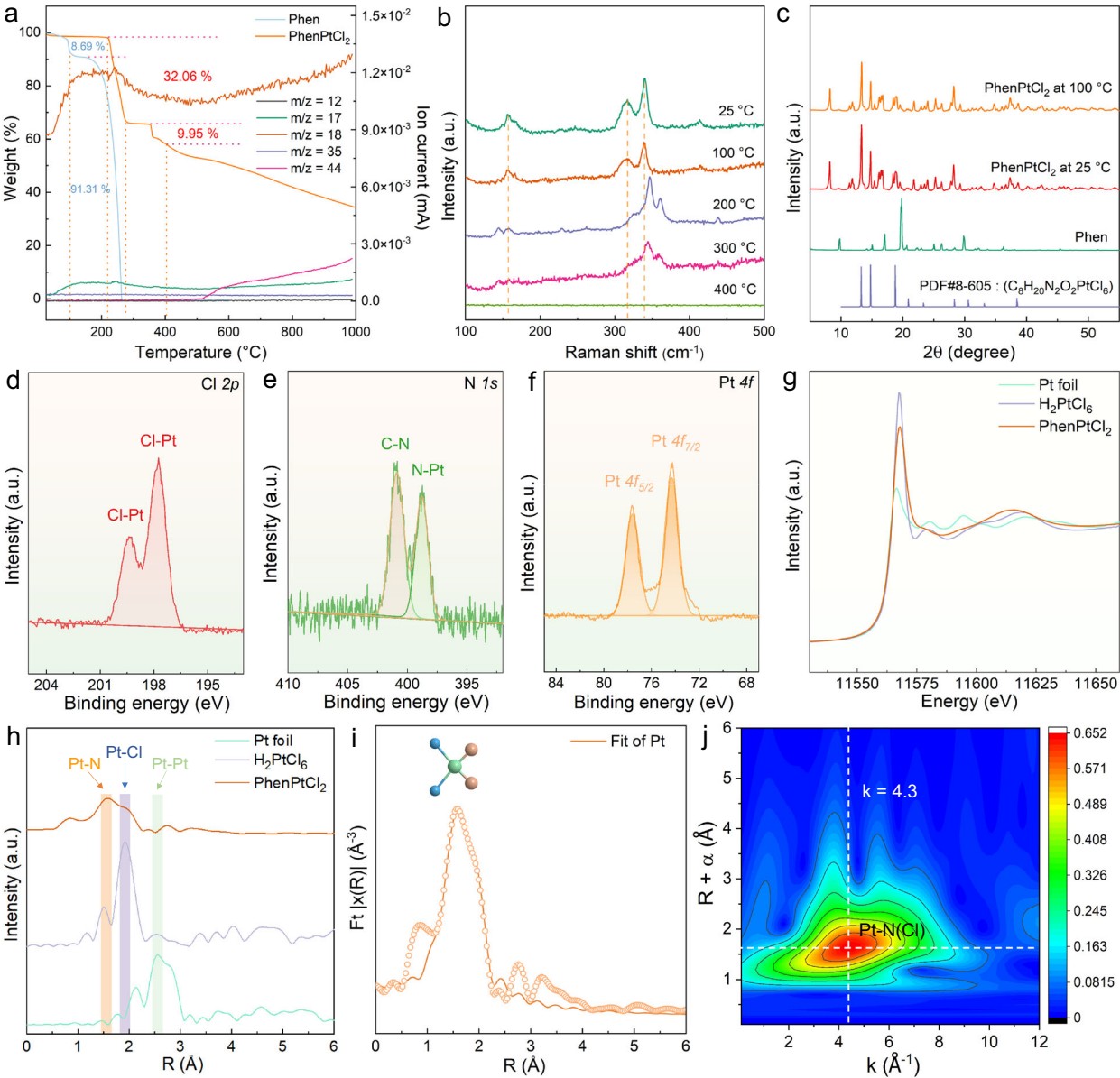

**Fig. 2 | Structural analysis of the 2D PhenPtCl₂ crystal. a** TG-MS of 2D PhenPtCl₂ crystal at 25 °C. **b** Raman spectroscopy of 2D PhenPtCl₂ crystal from 25 to 400 °C. **c** XRD patterns of 2D PhenPtCl₂ crystal at 25 and 100 °C. Among them, the organic molecule Phen and C₈H₂₀N₂O₂PtCl₆ are used as a contrast. High-resolution XPS of 2D PhenPtCl₂ crystal at 100 °C for (**d**) Cl *2p*, (**e**) N *1s* and (**f**) Pt *4f*. **g** Normalized XANES and (**h**) EXAFS spectra at the Pt L₃-edge for 2D PhenPtCl₂ nanosheets at 100 °C. Pt foil and H₂PtCl₆ are used as references. **i** Experimental and fitting EXAFS results of 2D PhenPtCl₂ nanosheets at 100 °C. The experimental and fitting results are shown as circles and solid lines, respectively. The inset atomic models are the first-shell coordination of Pt. **j** The WT for the Pt atom based on EXAFS signals of 2D PhenPtCl₂ nanosheets at 100 °C.

Subsequently, Raman spectroscopy was utilized to analyze the changes in the structural vibration peak of the PhenPtCl₂ crystal at various temperatures from 25 to 400 °C (Fig. 2b). Additionally, photographs of the organic crystals at different annealing temperatures from 25 to 400 °C show a color transformation from yellow to light yellow, and later from dark brown to black due to the changes in the light reflection properties (Supplementary Fig. 11). The optical images of the corresponding crystal powders exhibit a distinct lamellar structure. When the annealing temperature reaches 300 °C, the crystal displays significant decomposition, leading to the collapse of the lamellar arrangement (Supplementary Fig. 12). At 25 and 100 °C, the same Raman peak was observed, whereas the Raman peaks at 200 and 300 °C show similarity but were distinct from those at 25 and 100 °C. Notably, no Raman peaks were detected at 400 °C, indicating that the PhenPtCl₂ crystal has already initiated decomposition, transforming

into an amorphous carbon structure (Raman peak from 25 to 400 °C in Supplementary Fig. 13). Meanwhile, the crystal structure of 2D PhenPtCl₂ crystal at 25 and 100 °C was further revealed by X-ray diffraction (XRD), showing a completely different diffraction peaks compared with that of the Phen crystal, but a similar diffraction peak of the C₈H₂₀N₂O₂PtCl₆ (PDF#8-605) crystal (Fig. 2c)[30,31]. More importantly, TG-MS, XRD and Raman further prove that the 2D PhenPtCl₂ crystal is the same substance at 25 and 100 °C, with no structural evolution and decomposition. To further characterize the 2D PhenPtCl₂ crystal at 100 °C, iDPC-STEM imaging and EDS elemental maps were used to confirm no obvious structural difference from the 2D PhenPtCl₂ crystal at 25 °C (Supplementary Fig. 14).

Meanwhile, XPS was employed to analyze the elemental composition and valence states of 2D PhenPtCl₂ crystal at 100 °C, confirming the presence of C, N, Cl, and Pt elements as expected (Fig. 2d–f). The

detailed XPS discloses the specific peaks that provide insights into the bonding arrangements within the crystal. The Cl *2p* peaks at 197.7 and 199.3 eV indicate the existence of Cl-Pt bonding (Fig. 2d). The XPS peak of N *1s* appears at 398.8 eV, corresponding to the N-Pt bonding, and 400.9 eV for the C-N bonding (Fig. 2e). Notably, the valence states of Pt are also analyzed, revealing the binding energy of Pt *4f_{7/2}* is 74.3 eV, that of Pt *4f_{5/2}* is 77.6 eV (Fig. 2f). Among them, the peak value of Pt *4f_{7/2}* is close to 74.9 eV ($PtO_2$), and is far from 72.4 eV (PtO). This observation firmly supports the conclusion that Pt is engaged in a coordination environment characterized by a 4-coordination geometry. To ensure the structural stability of the 2D samples in the organic solvent, XPS was also conducted after the crystals were washed with ethanol, and no significant structural changes were observed (Supplementary Fig. 15). The XPS for the 2D $PhenPtCl_2$ samples at 25 and 100 °C was also analyzed, and no difference was observed (Supplementary Fig. 16). Furthermore, energy-dispersive X-ray spectroscopy (EDS) further confirms the presence of C, N, Cl, and Pt elements (Supplementary Fig. 17). Collectively, these results support the conclusion that the 2D $PhenPtCl_2$ crystal is primarily composed of four-coordinate species.

Furthermore, we conducted a comprehensive analysis of the coordination structure of Pt with two N atoms and two Cl atoms by using XAS, including X-ray absorption near-edge structure (XANES) and extended X-ray absorption fine structure (EXAFS) measurements. Among them, XANES measurements are used to investigate the electronic state of the Pt species at the L_3-edge, revealing the lower intensity of $PhenPtCl_2$ than that of $H_2PtCl_6$, but higher than that of the Pt foil (Fig. 2g)[32]. This suggests that Pt is positively charged with origin from Pt-N or Pt-Cl bonds. Figure 2h exhibits the Fourier transforms of the Pt L_3-edge EXAFS oscillations of the 2D $PhenPtCl_2$ crystal, in which the primary prominent shell is located at 1.56 Å, corresponding mainly to Pt-N bonds. Additionally, an acromion is detected at 1.93 Å, conforming to the Pt-Cl bonds. By fitting the Fourier transformed EXAFS curves, a Pt-N coordination number of 2.3 ± 0.2 and a Pt-Cl coordination number of 2.0 ± 0.3 were determined, which signifies that each Pt coordinates with two N and two Cl, forming a $N_2$-Pt-$Cl_2$ coordination structure (Fig. 2i and Supplementary Table 1). Based on the above analysis, each Pt atom in the $PhenPtCl_2$ crystal at 100 °C is stabilized by forming two Pt-N bonds and two Pt-Cl bonds (Set of Fig. 2i). Additionally, wavelet transforms EXAFS (WT-EXAFS) results further validate this coordination pattern, as the Pt-N(Cl) bond with a lower *k* value about 4.3 for the $PhenPtCl_2$ crystal, compared with $k = 8.1$ for Pt foil and $k = 6.0$ for $H_2PtCl_6$ powder (Fig. 2j and Supplementary Fig. 18). These findings provide robust evidence that Pt coordinates with two N and two Cl in the 2D $PhenPtCl_2$ crystal structure, supporting its unique coordination environment.

## Electrocatalytic HER performance of 2D PhenPtCl₂ nanosheets

The structural evolution of 2D $PhenPtCl_2$ under different annealing temperatures inevitably leads to the difference in HER performance. Therefore, the HER performance of the nanosheets was evaluated at different annealing temperatures across the full pH range (the corresponding catalyst inks as depicted in Supplementary Fig. 19). Remarkably, the 2D $PhenPtCl_2$ nanosheets exhibited excellent electrocatalytic hydrogen production, surpassing that of the commercial Pt/C in acidic, neutral, and alkaline solutions (Fig. 3a–e). Additional data are available for a detailed evaluation of HER performance in acidic, neutral, and alkaline solutions in Supplementary Figs. 20–22. Meanwhile, the 2D $PhenPtCl_2$ at 100 °C sample showed the largest electrochemical active surface area (ECSA) with 2.15 mF cm$^{-2}$ and the smallest transmission resistance with 32 Ω in acidic solution (Supplementary Fig. 20). After 36,000-s stability tests, the polarization curve of 2D $PhenPtCl_2$ nanosheets annealed at 100 °C still maintains no attenuation (Fig. 3c). Notably, 2D $PhenPtCl_2$ nanosheets annealed at 100 °C exhibit remarkable performance with a Tafel slope of

28.2 mV dec$^{-1}$ and an overpotential of 41 mV at 10 mA cm$^{-2}$ in the acidic solution (Fig. 3a, b), a Tafel slope of 212.3 mV dec$^{-1}$ and an overpotential of 399.6 mV in the neutral solution (Fig. 3d), and a Tafel slope of 70.6 mV dec$^{-1}$ and an overpotential of 13.0 mV in the alkaline solution (Fig. 3e). The extremely low overpotentials at 10 mA cm$^{-2}$ in acidic, neutral, and alkaline solutions are also confirmed by the overpotential comparison with the commercial Pt/C (Fig. 3f). These findings highlight the critical role of the 2D $PhenPtCl_2$ catalyst in enhancing the electrocatalytic performance, making them promising candidates for efficient HER in various pH environments.

More intriguingly, with increasing the annealing temperature of 2D $PhenPtCl_2$ nanosheets, the hydrogen evolution performance shows a rapid collapse upon the samples above 300 °C. The significant change in HER performance of $PhenPtCl_2$ can be directly attributed to the variation in its crystal structure. In particular, the crystal structure of 2D $PhenPtCl_2$ gradually decomposes, leading to a shift in the coordination structure of Pt from Phen-Pt-$Cl_2$ to amorphous materials, which is also confirmed by the Raman spectra of 2D $PhenPtCl_2$ nanosheets at different annealing temperatures. Meanwhile, the stability of the $PhenPtCl_2$ catalyst is also an important indicator for the evaluation of efficient catalysts, and the stability of the catalyst across the full pH range was evaluated. The results revealed that the catalyst displayed better stability in acid solution but slightly lower stability in neutral solution (Fig. 3g). The iDPC-STEM of the 2D $PhenPtCl_2$ crystal at 100 °C after stability test end in 2 h was conducted, and the catalysts still maintained relatively good crystallinity without large damage (Supplementary Fig. 23). When compared with other excellent catalysts reported thus far[8,33–42], the catalysts in this study represent excellent performance in acid solution (Fig. 3h). This highlights the remarkable potential of 2D $PhenPtCl_2$ nanosheets as highly efficient and stable catalysts, with promising applications in diverse fields requiring efficient hydrogen evolution.

## In situ characterization and catalytic mechanism of 2D PhenPtCl₂ nanosheets

To further understand the HER performance and the inherent catalytic mechanism of 2D $PhenPtCl_2$ crystals in the electrolytes, multiple in situ characterizations were adopted to reflect the structural evolution during the catalytic process, especially the changes in the coordination structure of Pt. In situ Raman spectroscopy was used to disclose the coordination structure evolution of the 2D $PhenPtCl_2$ catalyst at 100 °C in the acid electrolyte (Fig. 4a). During the 10-h durability test, the Raman vibration peak intensity of the Pt-Cl bond ($I_{Pt-Cl}$) and that of the Pt-N bond ($I_{Pt-N}$) in 2D $PhenPtCl_2$ crystal gradually decreased. Meanwhile, the ratio of $I_{Pt-Cl}/I_{Pt-N}$ gradually decreases along with the continuous electrocatalysis reaction (Fig. 4b), which strongly prove the dissociation of Cl⁻ in the catalytic process, and provides a clear explanation for the dynamic coordination of Cl⁻ throughout the catalytic reaction. In addition, a silver nitrate solution was employed to identify the presence of Cl⁻ ions in the electrolyte after the electrocatalytic reaction. To begin the process, barium nitrate solution was used to remove $SO_4^{2-}$ ions from the electrolyte, ensuring the accurate detection of Cl⁻ ions. Subsequently, the silver nitrate solution was introduced to titrate the Cl⁻ ions. During this titration process, the addition of silver nitrate results in the formation of white precipitates, strongly indicating the presence of Cl⁻ ions in the solution (Supplementary Fig. 24). This presence of Cl⁻ ions can be attributed to the loss of Cl ligands from the catalyst during the electrocatalytic process, and these Cl⁻ ions subsequently dissolve into the electrolyte. The identification of Cl⁻ ions in the electrolyte further confirms the dynamic coordination changes of the 2D $PhenPtCl_2$ nanosheets during the HER. Based on these conclusions, it is revealed that the $PhenPtCl_2$ crystal structure gradually loses Cl ligands during the entire electrocatalysis process, causing the change in the coordination number of Pt into an unsaturated coordination.

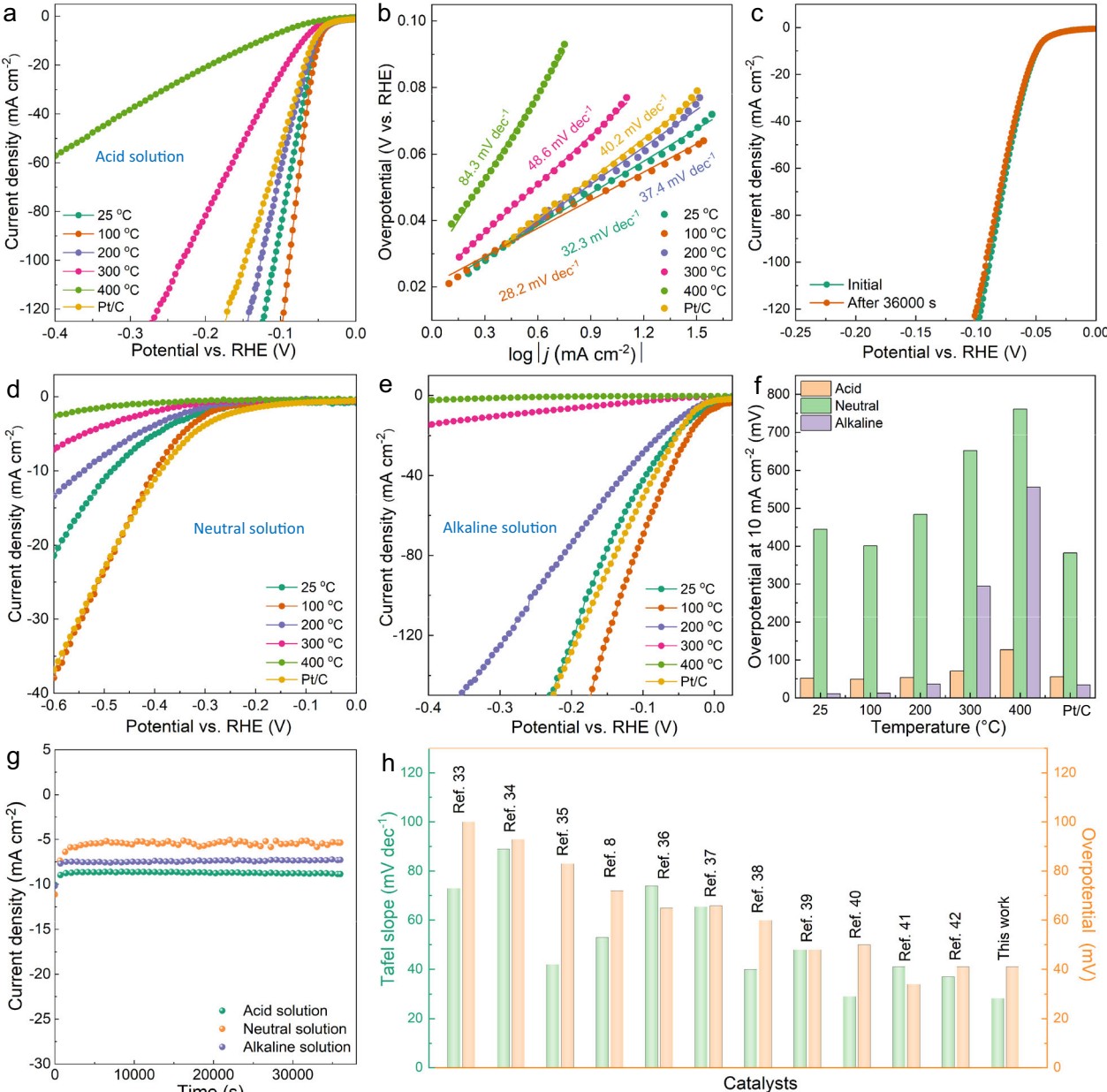

**Fig. 3 | HER performance evaluation across the full pH range of the 2D PhenPtCl₂ crystal. a** Cathodic polarization curves of 2D PhenPtCl₂ nanosheets under different annealing temperatures in acid solution (pH = 0.01). **b** Corresponding Tafel slopes derived from polarization curves in acid solution. **c** Polarization curve before and after the 36,000 s stability test. Cathodic polarization curves of 2D PhenPtCl₂ nanosheets under different annealing temperatures in (**d**) neutral solution (pH = 7.2) and (**e**) alkaline solution (pH = 14.01).

**f** Overpotential at 10 mA cm⁻² comparison of 2D PhenPtCl₂ nanosheets under different annealing temperatures compared to commercial Pt/C. **g** Durability test of 2D PhenPtCl₂ nanosheets at 100 °C at 10 mA cm⁻² in acid, neutral, and alkaline solution. **h** Comparison of the contrast samples for overpotential and Tafel slope[8,33–42]. In all the HER test, the catalyst mass loading is 0.57 mg cm⁻² on the working electrode. The resistance of catalyst for 32 Ω, 121 Ω and 56 Ω in acid, neutral, and alkaline solution, respectively.

To gain deeper insights into the dynamic interaction between Pt, Phen ligands, and Cl⁻ ions in the surrounding electrolyte, especially Cl⁻ ions, in situ XPS was used to analyze the evolution of Pt valence and the content change of Cl throughout the entire electrocatalytic process. From the XPS Pt *4f* peaks, we observed a rapid shift in peak positions to lower binding energies during electrocatalysis, followed by stabilization at a slightly higher energy level at the end of the HER. This trend indicates that the Pt valence decreases from +4 to +2 and eventually stabilizes at +3 after the HER ends. This transition in the valence state of Pt signifies a dynamic evolution process in the Pt coordination structure. The presence of two-coordinate Pt underscores its pivotal role in enhancing the HER

performance (Fig. 4c). Meanwhile, the content of Cl⁻ based on the peak intensity of Cl *2p* in 2D PhenPtCl₂ crystal firstly decreased and then increased with 2 h after the HER end. This finding indicate that the active Pt center undergoes a dynamic equilibrium process with the departure of Cl⁻ (Fig. 4d). Although the valence state of Pt and the peak intensity of Cl *2p* do not return to the original state after the HER, which is because other cations (H⁺) and Pt are in a competitive process in the acid electrolyte, preventing Pt from continuing to coordinate with Cl⁻. Based on the evolution trend in the intensity of Cl 2p peak at 197.7 eV and binding energy of Pt *4f₇/₂* for 2D PhenPtCl₂ crystal during the whole catalytic process and 2 h after HER in the acidic electrolyte, the dissociation and re-coordination of Cl⁻ and

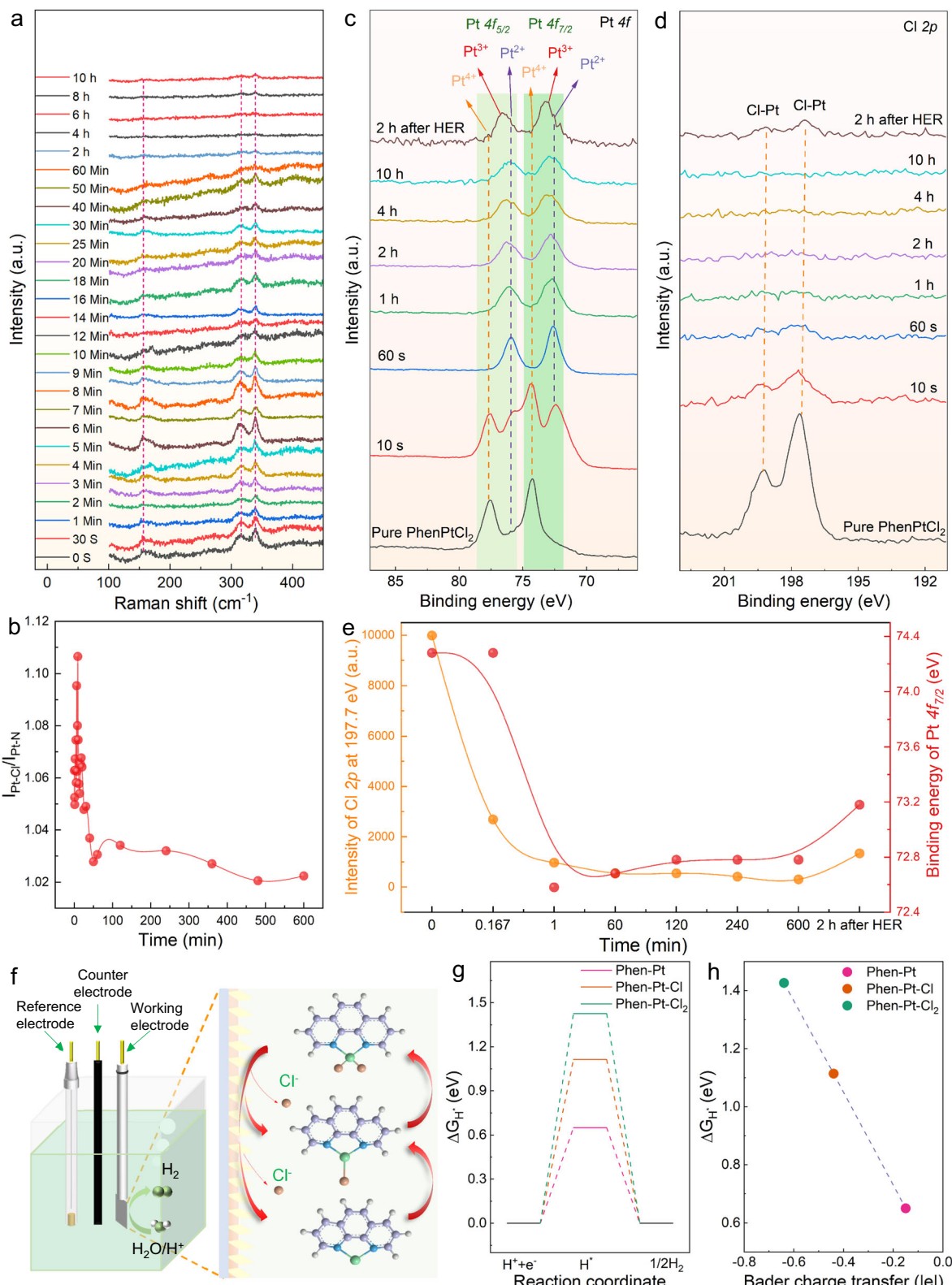

**Fig. 4 | Revelation of the catalytic mechanism in structural evolution of 2D PhenPtCl₂ crystal at 100 °C. a** In situ Raman spectroscopy characterization of the long stability test process for 2D PhenPtCl₂ crystal in the acid electrolyte. **b** Intensity ratio of $I_{Pt\text{-}Cl}/I_{Pt\text{-}N}$ for 2D PhenPtCl₂ crystal in the HER process. In situ XPS spectroscopy at (**c**) Pt *4f* peak and (**d**) Cl *2p* peak of 2D PhenPtCl₂ crystal during the whole HER process and 2 h after HER in the acid electrolyte. **e** Evolution trend in the

intensity of Cl *2p* peak at 197.7 eV and binding energy of Pt *4f₇/₂* for 2D PhenPtCl₂ crystal during the whole HER process and 2 h after HER in the acid electrolyte. **f** Schematic diagram of crystal structure evolution for 2D PhenPtCl₂ during HER process. **g** Calculated hydrogen adsorption free energy for Phen-Pt-Cl₂, Phen-Pt-Cl and Phen-Pt. **h** Relationship between Bader charge transfer and ΔG$_{H^*}$.

the dynamic change in the valence state of Pt in the HER are clearly indicated (Fig. 4e). These observations strongly suggest that Cl⁻ ions vacate the structure during the electrocatalysis process and subsequently re-coordinate with Pt after the HER. This provides direct evidence that the removal and retention of Cl⁻ in 2D PhenPtCl₂ samples drive the dynamic changes in Pt coordination structure. The active Pt site undergoes dynamic coordination evolution during the electrocatalytic process, with the unsaturated, two-coordinated Pt playing a crucial role in enhancing HER performance.

Meanwhile, the valence state of Pt and the content of Cl in the neutral and alkaline electrolyte were also tested by in situ XPS during the electrocatalytic HER process (Supplementary Fig. 25). The observed changes in the Cl content and Pt valence state are consistent with the patterns observed in acidic electrolytes. These results all indicate that the active Pt center in 2D PhenPtCl₂ crystal at 100 °C is in the dynamic coordination equilibrium during the HER process. Among then, it is this dynamic unsaturated coordination structure of Pt in the intermediate state, Phen-Pt that underpins the exceptional electrocatalytic performance of the 2D PhenPtCl₂ crystal, underscoring the intricate interplay between structural dynamics and electrocatalytic activity.

The shift in the valence state of Pt and the loss of Cl⁻ in the PhenPtCl₂ crystal from in situ XPS directly confirm the evolution of the Pt coordination structure, supporting the conclusion that the PhenPtCl₂ crystal structure undergoes dynamic changes during the electrocatalytic process. This crucial information contributes to a comprehensive understanding of the catalytic mechanism and the stability of the 2D PhenPtCl₂ nanosheets during hydrogen evolution. Based on the in situ Raman and in situ XPS, the evolution of the 2D PhenPtCl₂ crystal structure during the entire electrocatalytic process is inferred to undergo a series of structural changes, evolving from Phen-Pt-Cl₂ to Phen-Pt-Cl and into Phen-Pt (Fig. 4f). Among them, initially, the 2D PhenPtCl₂ nanosheets possess a crystal structure with coordination between Phen, Pt, and Cl⁻ ligands. This is the starting point of the electrocatalytic process. As the catalytic process proceeds, the 2D PhenPtCl₂ crystal begins to lose the Cl⁻ ligand, resulting in the formation of a Phen-Pt-Cl coordination structure. Furthermore, during the electrocatalytic process, the 2D PhenPtCl₂ nanosheets continue to evolve, leading to the complete loss of the Cl⁻ ligand and the formation of 2-coordination Pt with Phen. Therefore, the dynamic equilibrium allows the Phen-Pt structure to be the core catalytic state of the 2D crystal, with Pt coordinating primarily with Phen ligands and dynamically interacting with Cl⁻ ions in the surrounding electrolyte.

Density functional theory (DFT) calculations were performed to better understand the mechanism of HER on 2D PhenPtCl₂. The 2D planes of Phen-Pt-Cl₂, Phen-Pt-Cl and Phen-Pt were adopted to ensure that Pt is exposed to the vacuum region (Supplementary Fig. 26). For all these structures, the hydrogen adsorption site was optimized to calculate the free energy change of hydrogen adsorption ($\Delta G_{H^*}$). With the loss of Cl⁻, $\Delta G_{H^*}$ decreases significantly, and for Phen-Pt, $\Delta G_{H^*}$ reaches the lowest value, suggesting the best HER performance in Phen-Pt (Fig. 4g). Bader charge calculation was performed to explore the reason for the reduction in $\Delta G_{H^*}$ (Fig. 4h). The Bader charge transfer of Pt in Phen-Pt-Cl₂, Phen-Pt-Cl, and Phen-Pt shows similar trend to $\Delta G_{H^*}$. When the Bader charge transfer is less negative, i.e., Pt loses fewer electrons, $\Delta G_{H^*}$ becomes less negative, indicating higher HER activity. Naturally, the electronegative Cl withdraws electrons from Pt and causes the reduction of electrons around Pt, which is not beneficial to the Pt-H bonding. Compared with Phen-Pt-Cl₂ and Phen-Pt-Cl, Phen-Pt without Cl⁻ coordination means an optimal state for hydrogen adsorption and HER. Therefore, the Phen-Pt-Cl or Phen-Pt intermediates structures serve as the key catalytically active sites in the HER process, as the provides the dynamic coordination between Pt and Cl⁻ ions while maintaining the central coordination with Phen

ligands. This dynamic equilibrium is crucial in driving the efficient HER in 2D PhenPtCl₂ nanosheets.

## Discussion

In summary, this study focuses on the synthesis, electrocatalytic HER performance, and mechanism study of 2D PhenPtCl₂ nanosheets. Advanced iDPC-STEM technology reveals the crystal structure of 2D PhenPtCl₂ nanosheets with two Pt-N bonds and two Pt-Cl bonds. This 2D crystal with Phen-Pt-Cl₂ coordination structure, exhibits excellent HER performance across the full pH range. Through in-depth characterization with in situ Raman and in situ XPS, the Pt active sites of 2D PhenPtCl₂ nanosheets exhibit dynamic coordination evolution during the electrocatalytic process, changing from Phen-Pt-Cl₂ to Phen-Pt-Cl and finally to Phen-Pt. The PhenPt intermediate plays a central electrocatalytic role, dynamically coordinating with Cl⁻ ions in the electrolyte. The two-coordinated Pt in Phen-Pt provides more space and electrons, which facilitate H⁺ adsorption and H₂ evolution. The study sheds light on the intricate coordination chemistry and structural adaptability of nanosheets, offering promising prospects for efficient and tunable electrocatalysts in energy conversion and storage applications.

## Methods

### Materials and reagents
1,10-Phenanthroline ($C_{12}H_8N_2$), chloroplatinic acid ($H_2PtCl_6$), phosphate buffer saline (PBS), silver nitrate ($AgNO_3$), barium nitrate ($Ba(NO_3)_2$), KOH and ethanol were purchased from the Macklin Company, and used in experiments without further purification.

### Synthesis of 2D PhenPtCl₂ nanosheets
The 2D PhenPtCl₂ molecule crystal nanosheets were synthesized via the solvothermal method in the solution. Ultrasonic assistance can be added or not, if added, the size of sample is smaller. Among them, 180.2 mg of 1,10-Phenanthroline and 204.9 mg of chloroplatinic acid ($H_2PtCl_6$) were respectively added into 50 ml of deionized water or ethanol in the beaker. After the two precursors were dissolved in water or ethanol, and then mixed, the auxiliary ultrasound was used to react, resulting in a yellowish powder particle settling at the bottom of the beaker. Then the products were centrifugally washed three times with water and ethanol respectively, and freeze-dried to obtain yellow solid powder.

### Annealing treatment of 2D PhenPtCl₂ sample
To initiate the annealing process, the dried 2D PhenPtCl₂ crystal powders were meticulously distributed within a pristine quartz boat. Subsequently, this quartz boat was carefully positioned at the center of a quartz tube, part of a tube furnace equipped with a single temperature zone. The procedure commenced with the introduction of a substantial flow of 600 sccm Ar, serving the purpose of a 10-min ventilation to clean the furnace tube of any impurity gases. Thereafter, the gas composition was switched to a 100 sccm H₂/Ar mixture, containing 5% H₂. In this controlled atmosphere, the tube furnace was methodically heated to the predetermined temperature range (100, 200, 300, and 400 °C) for an elapsed time of 60 min. Then, the temperature was preserved for an additional 120 min to ensure annealing of all materials. The tube furnace then underwent a natural cooling process, gradually down to room temperature. The annealed sample was then extracted for subsequent characterization and performance testing.

### Characterization
The surface morphology of 2D PhenPtCl₂ nanosheets were examined with SEM (Hitachi-S4800). Optical images were taken by optical microscope (SOPTOP CX40M). XRD measurements were performed on a Bruker Dimension Icon D8 Advance system using Cu $K_\alpha$ radiation

(40 kV, 40 mA). TGA data were collected by thermal analysis system (NETZSCH STA449 F3). Solid-state nuclear magnetic resonance was analyzed by superconducting (solid) nuclear magnetic resonance (Bruker AVANCE 400WB). Infrared spectra were gathered by FT-IR (Thermo Fisher Scientific nicolt Is50). Raman spectra were performed using a WITec alpha 300 R spectrometer with 532 nm laser excitation. The X ray photoelectron spectrometer (XPS) spectra of these samples were analyzed by ESCALAB 250Xi XPS equipped with a monochromatic Al $K_\alpha$ source ($\lambda = 1486.6$ eV). The adventitious C *1s* peak of -284.8 eV was used for charging corrections. HAADF-STEM images and energy dispersive X-ray spectroscopy (EDS) mapping were acquired by the FEI Titan Cubed Themis G2 300 with a probe corrector and a monochromator at 200 kV. Photoemission endstations BL11B in Shanghai Synchrotron Radiation Facility (SSRF) was used for the help in characterizations.

### Electrochemical measurements

LSV was conducted in a three-electrode cell configuration using a CHI 760E electrochemical station (CH Instruments) with a $H_2SO_4$ solution (0.5 mol l$^{-1}$), phosphate buffered saline or KOH solution (1 mol l$^{-1}$). A Hg/HgCl$_2$ or Hg/HgO$_2$ electrode was used as the reference electrode and a graphite rod counter electrode, respectively. Two mg of 2D PhenPtCl$_2$ nanosheet and 40 µl of Nafion solution (5 wt%) were dispersed in 960 µl of ethanol, and then ultrasonic treatment for 60 min for forming a homogeneous ink. In total, 20 µl of these above catalyst inks were then loaded onto a 3 mm diameter glassy carbon as working electrode. Before measurement, the electrolyte was bubbled with N$_2$ for ≈30 min. All the electrochemical measurements were iR-corrected. Cyclic voltammograms were conducted at room temperature using the same standard three-electrode setup with various scan rates (10, 20, 40, 80, 120, 160, 200, 240 and 280 mV s$^{-1}$). Nyquist plots of these catalysts nanosheets were measured over the frequency range from 0.1 Hz to 100 kHz. The electrocatalytic stability of these catalysts was evaluated at the constant current density of 10 mA cm$^{-2}$.

### Computational details

Spin-polarized DFT calculations were performed by Vienna ab initio simulation package (VASP)[43]. The Perdew-Burke-Ernzerhof (PBE) exchange-correlation functional of generalized gradient approximation (GGA) was used[44]. To describe the van der Waals interaction DFT-D3 method of Grimme et al. was employed[45,46]. For sampling the Brillouin zone, $2 \times 2 \times 1$ k-points was adopted. The cutoff energy was set as 450 eV. The vacuum region of all the cells was not less than 15 Å above the plane.

The free energy calculation was based on computational hydrogen electrode (CHE) model to describe the ($H^+ + e^-$) transfer at pH = 0[47,48]. $\Delta G$ was calculated as $\Delta G = \Delta E - T\Delta S + \Delta ZPE$ ($T = 298.15$ K), where $\Delta E$ is the reaction energy obtained from DFT calculations. $\Delta S$ is entropy correction. $\Delta ZPE$ is the correction in zero-point energy (ZPE). $\Delta S$ and $\Delta ZPE$ were both calculated by VASPKIT code[49].

### Data availability

All data supporting the findings in this study are available from the Source Data. Source data are provided with this paper.

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

## Acknowledgements

This work was supported by the National Natural Science Foundation of China (22205209, 52202373, 22376062, and U21A200972), China Postdoctoral Science Foundation (2022M722867), Key Research Project of Higher Education Institutions in Henan Province (23A530001), Henan Province Key Research and Promotion Project-Scientific and Technological Breakthroughs (232102230088), the Science and Technology Commission of Shanghai Municipality (22ZR1415700), Shanghai Rising-star Program (20QA1402400), and the Fundamental Research Funds for the Central Universities. The computations were performed at the National Supercomputing Center in Zhengzhou, China.

## Author contributions

G.S. conceived the research and performed the experiments. Z.M. and X.Z. contributed to the DFT simulation. J.X., S.D., J.L. and C.J. performed STEM characterizations. D.G., Y.L., W.D., M.W., K.H. and Y.Y. performed data analysis. G.S., Z.Z., S.L., S.D. and J.X. contributed to manuscript editing. All authors contributed to the general discussion.

## Competing interests

The authors declare no competing interests.
