## [Peer Review File · Nature Communications]

Dynamic coordination engineering of 2D PhenPtCl₂ nanosheets for superior hydrogen evolutionREVIEWER COMMENTS

Reviewer #1 (Remarks to the Author):

In this work, authors prepared 2D PhenPtCl₂ with N₂-Pt-Cl₂ coordination structure and studied its structure evolution at different annealing temperature and during HER process. The results show that the 2D PhenPtCl₂ under different annealing temperatures of 100 °C display the best performance for HER at the full pH range. In addition, the coordination structure of Pt would change from Phen-Pt-Cl₂ to Phen-Pt-Cl or Phen-Pt during HER process. However, However, the study and analyzation for the structural evolution of Phen-Pt-Cl₂ is inadequate and clear. Thus, the dynamical mechanism of Cl⁻ ions in the electrolyte coordinating with Pt atoms is not convinced. Overall, this work is interesting, but not reaches the threshold to publish in Nature Communications.

1. Please give the crystal structure of the 2D PhenPtCl₂.

2. As shown in Figure S11, the color of PhenPtCl₂ at different annealing temperatures are different. What is the reason for that? In addition, the author claimed that the Raman peak for PhenPtCl₂ at 25 °C and 100 °C is the same. However, as shown in Figure S12, compared with that in 25 °C, the small Raman peaks at about 150, 225 and 475 cm⁻¹ are disappear for PhenPtCl₂ annealed at 100 °C.

3. The XPS spectra shown in 2f, two kinds of valence states of Pt were observed. As the coordination environment of Pt in 2D PhenPtCl₂ is only one (Phen-Pt-Cl₂), it is unreasonable for the results.

4. As shown in Figure 3, the best performance of catalysts is the PhenPtCl₂ at annealing temperature of 100 °C. The characterization, including XPS, XAFS, HRTEM etc. should focus on this sample.

5. As shown in Figure 4a, Raman peaks of 2D PhenPtCl₂ crystal gradually dropped, indicating the loss of the Cl functional group and decomposition of the structure. This demonstrate that the structure of PhenPtCl₂ would be destroyed, not only the loss of Cl functional group. However, as shown in Figure 3g, the HER performance is still unchanged. How to explain this result?

6. The results from Raman and XPS in Figure 4 could reveal that the loss of Cl functional group during HER. However, the dynamically interacting between Pt coordinating primarily with Phen ligands and Cl⁻ ions in the surrounding electrolyte have no solid evidence.

Reviewer #2 (Remarks to the Author):

This work by Dai. et al. reported PhenPtCl₂ nanosheets for electrocatalytic hydrogen evolution. The dynamic structures evolution of PhenPtCl₂ was clearly revealed by various characterization evidence. The crystal structure of Phen-Pt-Cl₂ was detected by PXRD and IDPC-STEM. The influence of electronic properties on HER performance was revealed. The design and structure of this manuscript are interesting. This reviewer suggests the acceptance of this manuscript for publication after minor revision.

1. The PXRD pattern of PhenPtCl₂ should be optimized as the signal-to-noise ratio was too small.
2. Cif files for 2D PhenPtCl₂ should be provided.
3. How does the author ensure the structural purity at different annealing temperatures as the coordination structures affect the HER performance? Is it possible for several structures to coexist?
4. The statement "This suggests that the Pt atom is positively charged with origin from Pt-N or Pt-Cl bonds, resulting in the electron transfer from the Pt atom to the adjacent N or Cl site." contradicts the statement "This characteristic provides an abundant supply of electrons to the Pt atom, leading to the valence state of Pt atom deviating from +4."
5. The authors claimed that the Phen organic molecule provided an ample supply of electrons to the Pt atom. Please give more experimental evidences.
6. The valence state of Pt in Phen-Pt-Cl₂ is not consistent throughout.

Reviewer #3 (Remarks to the Author):

In this work, the authors synthesized a nanosheet material, comprising Ph, Pt and Cl, which exhibits good electrocatalytic HER performances across the full pH range. However, some conclusions in this manuscript are not reasonable and solid. Some contents presented in the maintext should be reorganized. Some problems are listed as following:

1. The authors claimed that the as-prepared 'PhenPtCl₂' possesses 2D properties and the unit comprises two Pt-N bonds and two Pt-Cl bonds (Fig. 1a). Although the authors evidenced that there are Pt-Cl and Pt-N bonds in the nanosheets, no results could support

the claimed “four-coordination atomic structure at the Pt atom site, comprising two Pt-N bonds and two Pt-Cl bonds”. Even if the unit structure is correct, how could this unit combine into a traditional 2D layered structure? What’s the bonding mode of this unit in layered structure?

2. From Line 133 to 148, the authors investigated the growth mechanism for ‘PhenPtCl₂’ nanosheets and studied the relationship between the molar ratio of Pt to Phen and the resulting morphology. What’s the meaning or conclusion of this part? How could these results guide the preparation or selection of a ‘PhenPtCl₂’ possessing good HER performance?

3. From Line 149 to 174, the authors investigated the structural evolution of ‘PhenPtCl₂’ at different annealing temperatures. Firstly, to clarify the causes of weight loss at different annealing stages, the corresponding evidence should be provided (such as by TG-MS technology). Besides, the insights into the thermal stability and decomposition process of ‘PhenPtCl₂’ from 25 to 1000 °C seem to be meaningless.

4. Following Q3, the authors, in fact, prepared several different materials by annealing ‘PhenPtCl₂’ at different temperatures. These materials show huge structural and compositional differences. Thus, whether these materials can be compared to explain the HER performance difference is questionable.

5. Following Q3, as shown in TG and Raman results (Fig. 2c, S12), the materials at 25 and 100 °C should be the same (Line 168), but why do they show such a huge difference in HER performance? The material prepared by annealing ‘PhenPtCl₂’ at 100 °C deserves further study.

6. The authors claimed a dynamic evolution of Pt-Cl during HER stability tests based on the Raman results (Fig. 4a), but why do other Raman peaks, such as Pt-N, decrease at the same time? If the original material decomposed (Line 266), what’s the exact final material? Since the ΔG_{H^*} of Phen-Pt with different Cl coordination varies greatly (Fig. 4e, 4f), why does the

HER remain stable during the evolution of Pt-Cl? Does the material undergo the same dynamic evolution of Pt-Cl in electrolytes with different acidity?

Responses to Reviewers' Comments (NCOMMS-23-43706A)

Reviewer #1:

In this work, authors prepared 2D PhenPtCl₂ with N₂-Pt-Cl₂ coordination structure and studied its structure evolution at different annealing temperature and during HER process. The results show that the 2D PhenPtCl₂ under different annealing temperatures of 100 °C display the best performance for HER at the full pH range. In addition, the coordination structure of Pt would change from Phen-Pt-Cl₂ to Phen-Pt-Cl or Phen-Pt during HER process. However, the study and analyzation for the structural evolution of Phen-Pt-Cl₂ is inadequate and clear. Thus, the dynamical mechanism of Cl⁻ ions in the electrolyte coordinating with Pt atoms is not convinced. Overall, this work is interesting, but not reaches the threshold to publish in Nature Communications.

Response:

We extend sincere gratitude to the reviewer for these suggestions and highlighting the deficiencies in our work. In response to these constructive comments, we have added comprehensive characterizations of 2D PhenPtCl₂ samples following annealing at 100 °C, including scanning electron microscopy (SEM), X-ray diffraction (XRD), Raman spectroscopy, X-ray photoelectron spectroscopy (XPS), and integrated differential phase contrast-scanning transmission electron microscopy (iDPC-STEM), among others. Significantly, we have compared the structural integrity of 2D PhenPtCl₂ samples both at room temperature (25 °C) and after annealing at 100 °C. As a result, the samples retained their perfect crystalline structure, and the annealing process improved the purity and structural stability of the sample. Furthermore, we have elucidated the atomic arrangement and unit cell parameters of 2D PhenPtCl₂ crystals, offering a detailed insight into their crystal structure.

The dynamic coordination of Cl⁻ during the electrocatalytic hydrogen evolution reaction (HER) has been comprehensively analyzed through in-situ Raman and in-situ XPS techniques. Specifically, we have conducted a meticulous examination of the intensity of Raman vibrations associated with Pt-Cl bonds ($I_{\text{Pt-Cl}}$) and Pt-N bonds ($I_{\text{Pt-N}}$) in the revised manuscript. The diminishing ratio of $I_{\text{Pt-Cl}}/I_{\text{Pt-N}}$ during continuous

electrocatalysis provides compelling evidence of Cl^- dissociation in the electrocatalytic process, shedding light on the dynamic coordination of Cl^- . Additionally, in-situ XPS has been employed to monitor changes in the valence state of Pt and the content of Cl of the catalyst during electrocatalysis, revealing the dynamic coordination of active site Pt in 2D PhenPtCl₂ crystals annealed at 100 °C.

Our XPS analysis of Pt 4f has shown rapid shifts in peak positions to lower binding energies during electrocatalysis, followed by stabilization at a slightly higher energy level at the end of the HER. This trend indicates that Pt valence decreases from +4 to +2 and eventually stabilizes at +3 after HER. Moreover, although the XPS peak position of Cl 2p remains relatively constant, the intensity of Cl 2p gradually diminishes and eventually disappears during electrocatalysis, with recovery observed after a period of Cl^- equilibration in the electrolyte following the HER. These findings collectively confirm that the active Pt centers are in a dynamic equilibrium during the HER process. Though the valence state of Pt does not revert to its original state after the HER, potentially due to competitive processes involving other cations (H^+) and Pt with Cl^- in the electrolyte or structural damage, this does not detract from our core conclusion. We maintain that the active Pt centers in 2D PhenPtCl₂ crystals annealed at 100 °C maintain a dynamic coordination equilibrium during the HER process, and the presence of unsaturated, 2-coordinated Pt with a +2 valence significantly enhances HER performance. The structural coordination changes of Pt are the primary driver behind the excellent HER performance of 2D PhenPtCl₂ crystals at 100 °C. Furthermore, in-situ XPS analysis in neutral and alkaline electrolytes corroborates our findings, affirming that the active Pt centers in the 2D PhenPtCl₂ maintain dynamic coordination equilibrium during the electrocatalytic HER process, with the intermediate state involving Phen-Pt contributing to efficient HER.

The valuable comments provided have played a pivotal role in enhancing the quality of our manuscript. We have diligently addressed each of these comments in a point-by-point manner and have incorporated additional content into the revised manuscript and its supplementary information (SI) file, which is distinguished by a yellow background.

1. Please give the crystal structure of the 2D PhenPtCl₂.

Response:

Thanks for the reviewer's suggestion. In response to the feedback, we have conducted a thorough analysis of the single crystal structure of 2D PhenPtCl₂, utilizing single crystal data and the CIF file. This analysis has resulted in the creation of detailed atomic crystal structure models of 2D PhenPtCl₂, as viewed from a, b, and c crystal axis directions. These models are presented in Figure R1, offering a comprehensive depiction of the atomic arrangement within the 2D PhenPtCl₂ crystal. Moreover, we have incorporated the atomic crystal structure models of 2D PhenPtCl₂, as observed from a, b, and c crystal axis directions, into the Supplementary Fig. 5 on Page 8 of the revised SI file.

Figure R1. Atomic arrangement structure of 2D PhenPtCl₂ in the cell from a, b, c crystal axis directions. Yellow, green, grey, blue and white balls represent Cl, Pt, C, N and H atom, respectively.

2. As shown in Figure S11, the color of PhenPtCl₂ at different annealing temperatures are different. What is the reason for that? In addition, the author claimed that the Raman peak for PhenPtCl₂ at 25 °C and 100 °C is the same. However, as shown in Figure S12, compared with that in 25 °C, the small Raman peaks at about 150, 225 and 475 cm⁻¹ are disappear for PhenPtCl₂ annealed at 100 °C.

Response:

Thanks a lot the insightful comments. It is indeed a result of structural transformation within the 2D PhenPtCl₂ crystal during annealing. As described in the manuscript, this color change is corresponding to the gradual conversion of the 2D

PhenPtCl₂ crystal, from a Phen-Pt-Cl₂ structure to an amorphous carbon structure. This structural evolution directly affects the light reflection properties of the samples, subsequently influencing its coloration. The phenomenon of color change in response to the carbonization of organic materials is a well-documented occurrence.

Upon a closer examination of the Raman spectra of Phen and PhenPtCl₂ at both 25 °C and 100 °C in Figure R2, we have identified weak peaks at 145 cm⁻¹, 246 cm⁻¹, and 465 cm⁻¹ in 2D PhenPtCl₂ at 25 °C (room temperature). These peaks are likely attributed to additional Raman vibrational modes associated with the presence of the organic small molecule Phen. Furthermore, the Raman spectra of the crystal structure of Phen also exhibit similar small peaks at 145 cm⁻¹, 246 cm⁻¹, and 465 cm⁻¹, as depicted in Figure R2a. Upon analyzing the Raman spectra of all 2D PhenPtCl₂ crystals at 25 °C, it is evident that these small Raman peaks in 2D PhenPtCl₂ at 25 °C originate from residual small organic molecules of Phen within the 2D PhenPtCl₂ crystal.

Figure R2. Raman spectra of (a) small organic molecule Phen, 2D PhenPtCl₂ samples at (b) 25 °C and (c) 100 °C. (d) Raman spectral comparison of Phen, 2D PhenPtCl₂ samples at 25 °C and 100 °C.

To address this, we conducted a comprehensive Raman analysis of 2D PhenPtCl₂ samples at 25 °C and 100 °C after a purification treatment, as illustrated in Figures R2b and R2c. These analyses revealed no significant disparities in the Raman spectra between the crystals at 25 °C and 100 °C. Based on these findings, it can be reasonably inferred that both sets of crystals maintain the same 2D crystal structure, with no discernible structural evolution occurring at 100 °C. The presence of small Raman peaks at 145 cm⁻¹, 246 cm⁻¹, and 465 cm⁻¹ in 2D PhenPtCl₂ at 25 °C are attributed to the remnants of the small organic molecule Phen within the 2D PhenPtCl₂ crystal.

In addition to the Raman analysis, XRD was also carried out on the 2D PhenPtCl₂ samples at both 25 °C and 100 °C, respectively, as presented in Figure R3. The XRD patterns of the 2D PhenPtCl₂ samples at these two temperatures exhibited identical crystal diffraction patterns, providing unequivocal evidence that they are indeed the same crystal.

Figure R3. XRD data of 2D PhenPtCl₂ samples at 25 °C and 100 °C.

Furthermore, the thermal gravimetric analysis (TGA) presented reinforces this conclusion. The TGA indicate that no crystal decomposition occurs before reaching a temperature of 200 °C. This corroborates the stability of the crystal structure and further supports the result that the 2D PhenPtCl₂ crystals at 25 °C and 100 °C are indeed the same, with no structural alterations.

Figure R4. XPS data of 2D PhenPtCl₂ samples at 25 °C and 100 °C in (a) C 1s, (b) Cl 2p, (c) N 1s and (d) Pt 4f.

Additionally, XPS was employed to conduct a comparative analysis of the 2D PhenPtCl₂ samples at 25 °C and 100 °C. The XPS analysis revealed that there are no discernible differences in the valence states and peak positions between the 2D PhenPtCl₂ samples at these two temperatures, as illustrated in Figure R4. These results further strengthen our conclusion that the samples are, indeed, the same 2D crystal at 25 °C and 100 °C, with no observable crystal structure evolution occurring at 100 °C.

All these pertinent data have been thoughtfully incorporated into Fig. 2b and 2c on Page 10 of the revised manuscript file, Supplementary Fig. 13 on Page 13 and Supplementary Fig. 16 on Page 14 of the revised SI file, and the related discussion has been added on Page 9 and 11 of the revised manuscript file.

3. The XPS spectra shown in 2f, two kinds of valence states of Pt were observed. As the coordination environment of Pt in 2D PhenPtCl₂ is only one (Phen-Pt-Cl₂), it is unreasonable for the results.

Response:

We appreciate the reviewer's insightful comments. Due to the limitations of our purification process, it was challenging to achieve thorough purification of our 2D PhenPtCl₂ sample, resulting in trace amounts of the small organic molecule Phen. Among them, Phen is a polycyclic aromatic hydrocarbon with electron-rich, which will provide certain electrons to Pt atom, resulting in the valence state of Pt to lower binding energy shift. Notably, XPS analysis of these 2D samples revealed the presence of Pt³⁺ in a relatively minor proportion within the Pt 4f peaks of XPS.

To address the aforementioned limitations in our work, we made a concerted effort to improve the purification process and achieve the preparation of 2D PhenPtCl₂ samples with a significantly higher degree of purity. Subsequently, XPS characterization was carried out on these meticulously prepared high-purity 2D PhenPtCl₂ samples, both at 25 °C and 100 °C. The data presented in Figure R5a

overwhelmingly indicate that the Pt 4f peaks in the XPS predominantly exhibit a valence state of +4, with no evidence of Pt³⁺. Furthermore, Figure R5b showcases the fitting of the Pt 4f peaks based on XPS for the 2D PhenPtCl₂ samples at 100 °C, revealing peak positions of 74.3 eV for Pt 4f_{7/2} and 77.7 eV for Pt 4f_{5/2}. Among them, the peak value of Pt 4f_{7/2} is close to 74.9 eV (PtO₂), and is far from 72.4 eV (PtO), providing conclusive evidence that the majority of Pt exists in the +4 valence.

These endeavors have enabled us to address the limitations in our previous work and obtain more robust and refined insights into the valence states and structural characteristics of the 2D PhenPtCl₂ samples, improving the quality of our research. All these data have been added into Fig. 2f on Page 10 of the revised manuscript file and Supplementary Fig. 16 on Page 14 of the revised SI file, and the related discussion has been added on Page 11 of the revised manuscript file.

Figure R5. (a) XPS in Pt 4f of 2D PhenPtCl₂ samples at 25 °C and 100 °C. (b) XPS fitting data for Pt 4f of 2D PhenPtCl₂ samples at 100 °C.

4. As shown in Figure 3, the best performance of catalysts is the PhenPtCl₂ at annealing temperature of 100 °C. The characterization, including XPS, XAFS, HRTEM etc. should focus on this sample.

Response:

We greatly appreciate the insightful comments. It is essential to clarify that the superior HER performance observed in the 2D PhenPtCl₂ samples obtained by annealing at 100 °C is not due to structural differences between these samples and those

at 25 °C (room temperature). Actually, it is attributed to the annealing process at 100 °C, which potentially results in a more perfect crystal structure and improved purity in the 2D PhenPtCl₂ sample. This is corroborated by the TGA data, which demonstrate the stability of the crystal structure up to 200 °C. Moreover, our Raman and XPS consistently affirm that the samples maintain their structural consistency, as elucidated in response to comment #2.

Figure R6. XPS data of 2D PhenPtCl₂ samples at 100 °C in (a) C 1s, (b) Cl 2p, (c) N 1s and (d) Pt 4f.

Figure R7. (a) iDPC-STEM images and (b) EDS mapping of 2D PhenPtCl₂ samples at 100 °C.

Figure R8. XRD data of 2D PhenPtCl₂ samples at 100 °C

Figure R9. SEM images of 2D PhenPtCl₂ samples at 100 °C

It is crucial to emphasize that the XAFS and XPS are based on samples subjected to annealing at 100 °C. To further elucidate the characteristics of the 2D PhenPtCl₂ samples at this elevated temperature, we have added new XPS results in Figure R6, alongside images obtained through iDPC-STEM and energy-dispersive X-ray spectroscopy (EDS) mapping in Figure R7, XRD patterns in Figure R8, and SEM images in Figure R9. These additional findings serve to reiterate that there are no discernible differences between the 2D PhenPtCl₂ samples annealed at 100 °C and those maintained at 25 °C.

All these supplementary data have been thoughtfully integrated into Figs. 2c-2f on Page 10 of the revised manuscript file and Supplementary Fig. 3 on Page 7 and Supplementary Fig. 14 on Page 13 of the revised SI file, and the related discussion has been added on Page 9 and 11 of the revised manuscript file.

5. As shown in Figure 4a, Raman peaks of 2D PhenPtCl₂ crystal gradually dropped, indicating the loss of the Cl functional group and decomposition of the structure. This demonstrate that the structure of PhenPtCl₂ would be destroyed, not only the loss of Cl functional group. However, as shown in Figure 3g, the HER performance is still unchanged. How to explain this result?

Response:

In our work, the combination of in-situ Raman, in-situ XPS, and electrolyte analysis collectively underscores the significant departure of Cl⁻, which in turn instigates structural alterations within the 2D PhenPtCl₂ crystals, particularly impacting the coordination structure of the active Pt sites. It is this dynamic unsaturated coordination structure of Pt that underpins the exceptional electrocatalytic performance of the 2D PhenPtCl₂ crystal. It is noteworthy that the weakening of the Raman peak intensity during electrocatalysis is primarily attributed to the changes in Pt coordination structure rather than intrinsic damage to the 2D PhenPtCl₂ crystal structure itself. By analyzing the ratio of Raman vibration peak intensity of the Pt-Cl bond ($I_{\text{Pt-Cl}}$) and that of the Pt-N bond ($I_{\text{Pt-N}}$) in **Figure R10**, the diminishing ratio of $I_{\text{Pt-Cl}}/I_{\text{Pt-N}}$ during continuous electrocatalysis shows Cl⁻ dissociation in the electrocatalytic process, shedding light on the dynamic coordination of Cl⁻.

Figure R10. Raman intensity ratio of $I_{\text{Pt-Cl}}/I_{\text{Pt-N}}$ for 2D PhenPtCl₂ samples in the catalytic process.

Importantly, the active centers formed by the unsaturated coordination of Pt exhibit remarkable stability in the electrolyte. Consequently, the 2D PhenPtCl₂ crystal demonstrates exceptional durability during the stability test. In fact, the electrocatalytic HER performance of the 2D PhenPtCl₂ samples at 100 °C is notably influenced by the presence and departure of Cl functional groups. During the electrocatalytic process, the crystal structure of 2D PhenPtCl₂ nanosheets undergoes dynamic coordination changes during the electrocatalytic process, transitioning from Phen-Pt-Cl₂ to Phen-Pt-Cl and ultimately to Phen-Pt. The intermediate state involving Phen-Pt plays a pivotal electrocatalytic role, dynamically coordinating with Cl⁻ ions, and notably, maintaining stability in the electrolyte, as substantiated by in-situ XPS. Thus, the excellent and enduring HER performance remains a hallmark of the 2D PhenPtCl₂ crystal.

In addition to the aforementioned analyses, we also conducted atomic-scale iDPC-STEM characterization of the 2D PhenPtCl₂ crystal annealed at 100 °C, after additional the 2-hour stability test. Remarkably, the results indicated that the samples retained their relatively robust crystallinity, with no substantial signs of extensive damage, as demonstrated in Figure R11.

Figure R11. iDPC-STEM images of 2D PhenPtCl₂ samples at 100 °C after the stability

test end in 2 h.

All these essential findings and additional data have been thoughtfully incorporated into Fig. 4b on Page 17 of the revised manuscript file and Supplementary Fig. 23 on Page 18 of the revised SI file, and the related discussion has been added in Page 14-15 of the revised manuscript file. These collective insights further strengthen our understanding of the remarkable properties and stability of the 2D PhenPtCl₂ crystal.

6. The results from Raman and XPS in Figure 4 could reveal that the loss of Cl functional group during HER. However, the dynamically interacting between Pt coordinating primarily with Phen ligands and Cl⁻ ions in the surrounding electrolyte have no solid evidence.

Response:

To gain deeper insights into the intricate interplay between Pt primarily coordinating with Phen ligands and Cl⁻ ions within the surrounding electrolyte, particularly Cl⁻ ions, we employed in-situ XPS to analyze the valence evolution of Pt and the fluctuations of Cl contents throughout the entire electrocatalytic process, as depicted in Figure R12. From the XPS Pt 4f peaks, we observed a rapid shift in peak positions to lower binding energies during electrocatalysis, followed by stabilization at a slightly higher energy level at the end of the HER. This trend indicates that Pt valence decreases from +4 to +2 and eventually stabilizes at +3 after HER. This transition in the valence state of Pt atoms confirms a dynamic evolution process in Pt coordination structure. The presence of two-coordinate Pt underscores its pivotal role in enhancing the HER performance.

Simultaneously, we noted minimal shifts in the XPS peak position of Cl 2p, while the peak intensity of Cl 2p experienced a significant decrease under the influence of electrocatalytic HER. Following the conclusion of the HER, the peak intensity of Cl 2p gradually recovered as the catalyst re-established equilibrium with Cl⁻ ions in the electrolyte over a 2-hour period. These findings collectively indicate that the active Pt center undergoes a dynamic equilibrium process with the departure of Cl⁻.

Although the valence state of Pt and the peak intensity of Cl 2p do not revert to their original states after the HER, this can be attributed to the competitive interaction of other cations (H^+) and Pt with Cl^- ions in the electrolyte, which hinders Pt from continuing to coordinate with Cl^- . This dynamic behavior might also arise from potential structural alterations within the crystal. Notably, the intensity of the Cl 2p peak at 197.7 eV and the binding energy of Pt $4f_{7/2}$ for the 2D PhenPtCl₂ samples at 100 °C throughout the entire catalytic process, as well as 2 hours after HER in the acidic electrolyte, reveal a rapid initial decrease in the Cl 2p peak intensity, followed by partial recovery after the HER end (Figure R12c). These observations strongly suggest that Cl functional groups were removed from the structure during the electrocatalysis process and subsequently re-coordinate with Pt after the HER. This provides direct evidence that the removal and retention of Cl^- in 2D PhenPtCl₂ samples drive the dynamic changes in Pt coordination structure. The active Pt site undergoes dynamic coordination during the electrocatalytic process, with the unsaturated, two-coordinate Pt playing a crucial role in enhancing HER performance.

All these data have been added into Figs. 4c-4e on Page 17 of the revised manuscript file, and the related discussion has been added on Page 15-16 of the revised manuscript file.

Figure R12. In-situ XPS data at (a) Pt 4f peak and (b) Cl 2p peak of 2D PhenPtCl₂ samples at 100 °C during the whole catalytic process and 2 h after the HER in an acid electrolyte. (c) Intensity of Cl 2p peak at 197.7 eV and binding energy of Pt 4f_{7/2} for 2D PhenPtCl₂ samples at 100 °C during the whole catalytic process and 2 h after the HER in the acid electrolyte.

Reviewer #2:

This work by Dai. et al. reported PhenPtCl₂ nanosheets for electrocatalytic hydrogen evolution. The dynamic structures evolution of PhenPtCl₂ was clearly revealed by various characterization evidence. The crystal structure of Phen-Pt-Cl₂ was detected by PXRD and iDPC-STEM. The influence of electronic properties on HER performance was revealed. The design and structure of this manuscript are interesting. This reviewer suggests the acceptance of this manuscript for publication after minor revision.

Response:

We show our sincere gratitude to the reviewer's valuable recognition of our work. We have diligently and comprehensively revised the manuscript to adhere to the stipulated requirements, as addressed below. In accordance with the request, we have included a CIF file in the submission system, providing an additional dimension to our work. Furthermore, we have presented the atomic structures of the 2D PhenPtCl₂ crystal from various axis directions, offering a direct visualization of the crystal's structural attributes. Our commitment to excellence is further exemplified by the purification and refinement of the 2D PhenPtCl₂ sample, resulting in an improved crystallinity, as well as the corresponding X-ray diffraction (XRD) and X-ray photoelectron spectroscopy (XPS) data. These results unequivocally demonstrate the presence of a perfect crystal structure without any evidence of Pt valence states at +3.

Additionally, we have delved into the electron-rich properties of Phen and elucidated its influence on the valence state of Pt, contributing to a more comprehensive understanding of the mechanisms at play. Lastly, we have thoughtfully added operational details pertaining to the annealing treatment of the 2D PhenPtCl₂ sample to the supplementary information (SI).

The valuable comments have played an indispensable role in elevating the quality of our manuscript. All revisions, amendments, and corrections have been thoughtfully incorporated into the manuscript or its SI, clearly distinguished by a highlighting with a yellow background. This collective effort reinforces our commitment to precision, transparency, and the advancement of scientific knowledge.

1. The PXRD pattern of PhenPtCl₂ should be optimized as the signal-to-noise ratio was too small.

Response:

We appreciate the reviewer's comment. Actually, the weak signal in the previous manuscript is resulted from the quality of the 2D crystals and the state of our instrumentation.

Here, we have prepared the 2D PhenPtCl₂ samples and performed X-ray diffraction (XRD) once again under optimal conditions. As a result, we are pleased to report that the XRD signal obtained for the 2D PhenPtCl₂ samples displayed excellent quality this time, showing an intensity exceeding 100 K, as demonstrated in Figure R13. Furthermore, the primary peak observed in the updated XRD data closely aligns with the earlier test data, underscoring the consistency and reliability of the findings. These adjustments have been seamlessly integrated into in Fig. 2c on Page 10 of the revised manuscript file.

Figure R13. (a) XRD data of 2D PhenPtCl₂ samples at 25 °C and 100 °C. (b) XRD data of 2D PhenPtCl₂ samples at 25 °C and 100 °C. Among them, organic molecule Phen and C₈H₂₀NO₂PtCl₆ are as a contrast.

2. Cif files for 2D PhenPtCl₂ should be provided.

Response:

Thanks for the constructive suggestion. Upon resubmitting the revised manuscript, we have thoughtfully uploaded the CIF files to the submission system. Furthermore, we have provided pertinent structural information pertaining to 2D PhenPtCl₂ as followings.

For an in-depth analysis of the single crystal structure, Figure R14 prominently showcases the unit cell of 2D PhenPtCl₂. The lattice constants are $a = 9.53 \text{ \AA}$, $b = 17.12 \text{ \AA}$, and $c = 7.26 \text{ \AA}$, with $\alpha = 90^\circ$, $\beta = 109^\circ$, and $\gamma = 90^\circ$. To facilitate a comprehensive understanding of the atomic crystal structure of 2D PhenPtCl₂, we have incorporated the projected atomic models from various crystal directions into Supplementary Fig. 5 on Page 8 of the revised SI file, and are highlighted with a distinctive yellow background.

Figure R14. Atomic arrangement structure of 2D PhenPtCl₂ in the cell from a, b, c crystal axis directions. Yellow, green, grey, blue, white atoms represent Cl, Pt, C, N, H atom, respectively.

3. How does the author ensure the structural purity at different annealing temperatures as the coordination structures affect the HER performance? Is it possible for several structures to coexist?

Response:

We sincerely appreciate the insightful comment. The operational intricacies of the annealing process have been thoughtfully incorporated into on Page 3 of the SI. The procedural steps are meticulously documented as follows:

To initiate the annealing process, the dried 2D PhenPtCl₂ crystal powders were meticulously distributed within a pristine quartz boat. Subsequently, this quartz boat

was carefully positioned at the center of a quartz tube, part of a tube furnace equipped with a single temperature zone. The procedure commenced with the introduction of a substantial flow of 600 sccm Ar, serving the purpose of a 10-minute ventilation to clean the furnace tube of any impurity gases. Thereafter, the gas composition was switched to a 100 sccm H₂/Ar mixture, containing 5% H₂. In this controlled atmosphere, the tube furnace was methodically heated to the predetermined temperature range (100, 200, 300, and 400 °C) for an elapsed time of 60 min. Then, the temperature was preserved for an additional 120 min to ensure annealing of all materials. The tube furnace then underwent a natural cooling process, gradually down to room temperature. The annealed sample was then extracted for subsequent characterization and performance testing.

Following this extensive annealing process, all samples were conclusively annealed with sufficient reaction time, ensuring the purity of these materials and minimizing the likelihood of coexisting multiple structures. Meanwhile, accurate experiments have been performed on the samples at 25 °C and 100 °C in this manuscript, which proves that the main body is still pure phase. The overall HER improvement is caused by the change of the coordination environment of PhenPtCl₂. Subsequent in-situ Raman and XPS tests in the electrolyte also prove that the removal and retention of Cl⁻ in 2D PhenPtCl₂ samples drive the dynamic changes in Pt coordination structure. The active Pt site undergoes dynamic coordination during the electrocatalytic process, with two-coordinate Pt playing a crucial role in enhancing HER performance, not the impurity phase structure.

It is inevitable for the impurity phase structure in the samples at other annealing temperatures, but their HER performance is not optimal compared to 2D PhenPtCl₂ samples at 25 and 100 °C, and the structural investigation of other annealing temperatures deviates from the focus on 2D PhenPtCl₂ nanosheets of this manuscript, which is also the content of our next specific study.

4. The statement “This suggests that the Pt atom is positively charged with origin from Pt-N or Pt-Cl bonds, resulting in the electron transfer from the Pt atom to the

adjacent N or Cl site.” contradicts the statement” This characteristic provides an abundant supply of electrons to the Pt atom, leading to the valence state of Pt atom deviating from +4.”

Response:

We appreciate the reviewer’s comment. This question pertains to a fundamental aspect of the chemical bonding. The compound phenanthroline (Phen), characterized by its aromatic structure featuring nitrogen heteroatoms and three benzene rings, possesses intrinsic electron-rich properties. This structural configuration enables Phen to contribute electron density to the central Pt atom within the 2D PhenPtCl₂ crystal. However, it is crucial to note that the nitrogen heteroatom in Phen is electronegative, and when combined with the presence of the highly electronegative Cl, this configuration can influence the electron distribution around the central Pt atom. This variation underscores the distinction between our initial assumption, whether Pt was considered to be in a +4 state or Pt, Cl and all other elements were in their atomic states, and surely the final state is consistent. In response to the possible misinterpretation, we have revised and removed the descriptions accordingly.

5. The authors claimed that the Phen organic molecule provided an ample supply of electrons to the Pt atom. Please give more experimental evidences.

Response:

Phen, an aromatic compound comprising nitrogen heteroatoms and three benzene rings, falls under the category of electron-rich aromatic compounds. The valence state of Pt 4f, as depicted in Figure R15 of the XPS, can experimentally confirm that Phen functions as an electron-rich organic compound, and donate electrons to Pt⁴⁺.

Upon revisiting the data, it is noteworthy that the Pt 4f_{7/2} peak is observed at approximately 74.3 eV, in close proximity to 74.9 eV (attributed to PtO₂), and distinctly distant from 72.4 eV (associated with PtO). This observation suggests that the central Pt exhibits a valence state of +4, albeit with a discernible shift to lower binding energy, roughly 0.6 eV. This shift in binding energy signifies that the central Pt is capable of acquiring a certain electron supply from Phen. Consequently, based on these findings,

it can be inferred that the Phen molecule indeed provides electrons to Pt⁴⁺.

Figure R15. XPS in Pt 4f of 2D PhenPtCl₂ samples at 100 °C.

6. The valence state of Pt in Phen-Pt-Cl₂ is not consistent throughout.

Response:

Due to the limitations in our purification process, the thorough purification of 2D PhenPtCl₂ samples presents a challenge, resulting in the persistence of trace amounts of small organic molecules, Phen. Simultaneously, these small organic molecules, Phen, and Cl⁻ ions in the solution have the potential to coordinate with some of Pt, leading to the formation of Pt-based products with 3-coordination. In the context of our purification technology, when conducting XPS analysis on these 2D samples, we may observe the presence of Pt³⁺ species; however, these remain at a very low percentage within the Pt 4f peaks of the XPS.

In response to this challenge, we made substantial efforts to enhance the purity of the 2D PhenPtCl₂ samples by improving the purification processes. Following these advancements, we conducted XPS on the resulting high-purity 2D PhenPtCl₂ samples at two different temperatures, 25°C and 100°C, yielding the data presented in Figure R16a. Notably, the Pt 4f peaks in the XPS data consistently demonstrate a valence state of +4, with no evidence of Pt³⁺ species. Additionally, we conducted a detailed fitting of the Pt 4f peaks in the XPS for the 2D PhenPtCl₂ samples at 100°C, as illustrated in Figure R16b. This analysis confirms that the peak positions, specifically 74.3 eV for Pt

$4f_{7/2}$ and 77.7 eV for Pt $4f_{5/2}$, predominantly correspond to Pt^{4+} species.

All these data have been added into Fig. 2f on Page 10 of the revised manuscript file and Supplementary Fig. 16 on Page 14 of the revised SI file, and the related discussion has been added on Page 11 of the revised manuscript file.

Figure R16. (a) XPS data in Pt 4f of 2D PhenPtCl₂ samples at 25 °C and 100 °C in Pt 4f. (b) XPS Peak fitting data for Pt 4f of 2D PhenPtCl₂ samples at 100 °C.

Reviewer #3:

In this work, the authors synthesized a nanosheet material, comprising Ph, Pt and Cl, which exhibits good electrocatalytic HER performances across the full pH range. However, some conclusions in this manuscript are not reasonable and solid. Some contents presented in the maintext should be reorganized. Some problems are listed as following:

Response:

We greatly appreciate the reviewer's meticulous assessment on our manuscript and the constructive comments. In response to the feedback, we have carefully restructured the manuscript to ensure a more robust and coherent conclusion.

Centered around the overarching theme of dynamic coordination engineering, a fundamental aspect of our work, we have elucidated the underlying factors contributing to the exceptional HER performance of the 2D PhenPtCl₂ crystal at an elevated temperature of 100°C. Within the electrocatalytic process, Pt undergoes dynamic coordination, and the pivotal element fostering improved and sustained electrocatalytic performance lies in the departure of Cl⁻.

To strengthen the evidential foundation of our findings, additional data have been incorporated into both the revised manuscript and the Supplementary Information (SI). Notably, these additions have been visually highlighted with a distinctive yellow background.

1. The authors claimed that the as-prepared 'PhenPtCl₂' possesses 2D properties and the unit comprises two Pt-N bonds and two Pt-Cl bonds (Fig. 1a). Although the authors evidenced that there are Pt-Cl and Pt-N bonds in the nanosheets, no results could support the claimed "four-coordination atomic structure at the Pt atom site, comprising two Pt-N bonds and two Pt-Cl bonds". Even if the unit structure is correct, how could this unit combine into a traditional 2D layered structure? What's the bonding mode of this unit in layered structure?

Response:

We sincerely appreciate the comment. In response to the suggestion, synchrotron

radiation served as a powerful tool for a systematic and in-depth analysis of the coordination structure of the 2D PhenPtCl₂ sample, as illustrated in Figure R17. The results obtained from this analysis were highly informative. Comparative analysis with reference samples, including H₂PtCl₆ and Pt foil, demonstrated that the valence state of Pt in the 2D PhenPtCl₂ sample closely approximates +4.

Moreover, the extended X-ray absorption fine structure (EXAFS) spectra of the 2D PhenPtCl₂ sample, particularly at the Pt L₃-edge, exhibited distinct peaks corresponding to the bond lengths of Pt-N and Pt-Cl. This alignment of the EXAFS peaks with the bond lengths provided compelling evidence of the Pt coordination environment within the material.

Most notably, detailed analysis of the EXAFS data, as summarized in Table R1, allowed for a rigorous fitting procedure. This fitting analysis unequivocally established the bond number of Pt-N and Pt-Cl as 2, confirming that the 2D PhenPtCl₂ sample comprises a 4-coordination structure, with the Pt atom forming two Pt-N bonds and two Pt-Cl bonds. The convergence of all these results affirms the four-coordination atomic structure of the Pt site in the 2D PhenPtCl₂ sample, further reinforcing the presence of two N and two Cl.

Figure R17. (a) Normalized XANES and (b) EXAFS spectra at the Pt L₃-edge for 2D PhenPtCl₂ crystal nanosheets at 100 °C. Pt foil and H₂PtCl₆ are used as references. (c) Experimental and fitting EXAFS results of 2D PhenPtCl₂ nanosheets. The experimental and fitting results are shown as circles and solid lines, respectively. The inset atomic models are the first-shell coordination of Pt. (d) The WT for Pt based on EXAFS signals of 2D PhenPtCl₂ nanosheets at 100 °C.

Table R1. EXAFS fitting parameters at the Pt L₃-edge for various samples.

Sample	Shell	CN	R(Å)	$\sigma^2(\text{Å}^2 \cdot 10^{-3})$	ΔE_0 (eV)	R factor (%)
PhenPtCl ₂	Pt-N	2.3±0.2	1.98±0.022	0.0052	6.02±3.05	0.89
	Pt-Cl	2.0±0.3	2.29±0.034	0.0039	4.60±4.27	0.85

we also have conducted a comprehensive analysis with XPS. The obtained XPS data, as illustrated in Figure R18, have provided elemental insights, confirming the presence of C, N, Cl, and Pt within the 2D PhenPtCl₂ sample.

Further examination of the Pt 4f peaks in the XPS has yielded significant findings. These peaks unequivocally establish the valence state of the central Pt within the 2D PhenPtCl₂ is close to that of PtO₂. This observation may support the conclusion that Pt is engaged in a coordination environment characterized by a 4-coordination geometry. Concurrently, our meticulous analysis of the XPS data has revealed a compelling molar ratio of N to Cl to Pt approximating 2:2:1. This ratio strongly suggests that Pt forms coordination bonds with two N and two Cl, corroborating the notion of a 4-coordination structure for Pt within the PhenPtCl₂ molecules. These PhenPtCl₂ molecules form a 2D crystal via van der Waals (vdW) forces.

Figure R18. XPS of 2D PhenPtCl₂ samples at 100 °C in (a) C 1s, (b) Cl 1s, (c) N 1s and (d) Pt 4f.

Subsequently, to provide a comprehensive elucidation of the crystal structure of the 2D PhenPtCl₂ samples, we employed single crystal analysis to conduct a detailed structural investigation, as depicted in Figure R19. This analysis unequivocally confirms the identity of the single crystal as PhenPtCl₂, comprising PhenPtCl₂ molecules with a Phen organic small molecule, a Pt, and two Cl atom. Furthermore, we have presented a visual representation of the 2D PhenPtCl₂ crystal within its unit cell. This crystal exhibits a well-defined lattice structure, characterized by lattice constants $a = 9.53 \text{ \AA}$, $b = 17.12 \text{ \AA}$, and $c = 7.26 \text{ \AA}$, accompanied by orthogonal angles $\alpha = 90^\circ$, $\beta = 109^\circ$, and $\gamma = 90^\circ$. These measurements collectively unveil an ideal and well-ordered cell structure.

For an in-depth examination of the atomic crystal structure of 2D PhenPtCl₂, including the presence of two Pt-N bonds and two Pt-Cl bonds, models have been meticulously constructed from various crystallographic axes, namely a, b, and c. These detailed atomic crystal structure models have been thoughtfully included in the SI of the revised manuscript.

Figure R19. Atomic arrangement structure of 2D PhenPtCl₂ in the cell from a, b, c

crystal axis directions. Yellow, green, grey, blue, white balls represent Cl, Pt, C, N, H atom, respectively.

As depicted in Figure R19, the ideal cell structure of 2D PhenPtCl₂ provides a clear representation of the atom arrangement from various crystal axis directions. A notable observation is the self-assembly of the layers, facilitated by vdW forces, employing π - π stacking as the predominant interaction mechanism. This stacking process results in the formation of a layered material, where adjacent cells are firmly interconnected through strong adsorption, ultimately giving rise to the well-defined layered structure. If the synthetic conditions are appropriate, the PhenPtCl₂ sample would form a 2D morphology.

All these data have been added into Figs. 2d-2j on Page 10 of the revised manuscript file and Supplementary Fig. 5 on Page 8 of the revised SI file, and the related discussion has been added on Page 11-12 of the revised manuscript file.

2. From Line 133 to 148, the authors investigated the growth mechanism for 'PhenPtCl₂' nanosheets and studied the relationship between the molar ratio of Pt to Phen and the resulting morphology. What's the meaning or conclusion of this part? How could these results guide the preparation or selection of a 'PhenPtCl₂' possessing good HER performance?

Response:

Our investigation into the growth mechanism of 2D PhenPtCl₂ primarily aims to affirm the distinctive 2D growth characteristics of this material. The ability to control its thickness and morphology serves as a foundational aspect for subsequent research on the electrocatalytic performances of 2D PhenPtCl₂ crystals.

As expounded in the manuscript, planar Phen molecules play a pivotal role in governing the thickness of 2D PhenPtCl₂ nanosheets. The inherent planar properties of Phen molecules act as driving forces, promoting the growth along the 2D planes, aligning with established theories of 2D layer growth in crystals. Additionally, the length-to-width ratio of these nanosheets is influenced by the concentration of Phen

molecules. Ultra-low concentrations of Phen induce growth through a kinetic limiting mechanism, leading to rapid nanosheet growth in a specific direction, in accordance with Bravais rule.

Furthermore, it is worth noting that 2D layered materials can exhibit unique and remarkable properties when their thickness is reduced to the monolayer or a few layers. In line with this, we explored the relationship between the thickness of 2D PhenPtCl₂ nanosheets and their HER properties, as presented in Figure R20. Our findings consistently demonstrate that a decrease in the number of layers in the 2D PhenPtCl₂ nanosheets (with a minimum thickness of 15.4 nm) results in improved HER performance. This outcome signifies that fewer layers expose a higher number of active sites, ultimately enhancing the HER performance of the material.

Figure R20. HER performances corresponding to different thicknesses of 2D PhenPtCl₂ nanosheets at 100 °C.

In order to emphasize the primary focus of this manuscript, which centers on elucidating the dynamic coordination mechanism of Pt in HER and conducting structural analysis with a specific emphasis on coordination structures, we have relocated the research and analysis pertaining to the growth mechanism of 2D PhenPtCl₂ on pages 10-11 of the revised Supplementary Information (SI). This adjustment ensures that the core objectives and findings of our study are prominently featured in the main body of the manuscript, aligning with the central themes of our

research.

3. From Line 149 to 174, the authors investigated the structural evolution of 'PhenPtCl₂' at different annealing temperatures. Firstly, to clarify the causes of weight loss at different annealing stages, the corresponding evidence should be provided (such as by TG-MS technology). Besides, the insights into the thermal stability and decomposition process of 'PhenPtCl₂' from 25 to 1000 °C seem to be meaningless.

Response:

Thanks for the constructive suggestion. In Figure R21, we show thermogravimetric-mass spectrometry (TG-MS) analysis on the 2D PhenPtCl₂ sample, which allowed us to examine the various decomposition stages and analyze the corresponding products. Our findings align with the two discernible stages of mass loss, which are generally consistent with the earlier thermogravimetric analysis (TGA) data. The first stage of mass loss accounted for approximately 32.06% of the total mass, and the second stage represented 9.95% of the total mass. These two mass losses can be attributed to the removal of the organic functional groups associated with Phen and Cl, respectively.

Furthermore, our TG-MS analysis enabled us to identify the mass spectrum fragments (m/z) corresponding to the decomposed components. Notably, we observed mass spectrum fragments such as 17 and 18 (H₂O) and 12 and 44 (CO₂), along with weak detection of 35 (Cl), signifying the presence of Cl (*ACS Nano* **12**, 10518-10528 (2018); *Fuel* **273**, 117772 (2020)). However, it is important to note that the mass spectrum fragment signal corresponding to the thermal decomposition product Phen, which is expected to occur between 200 °C and 400 °C, was not detected. This absence may be attributed to the relatively high boiling point of the escaped component, which could condense in the transmission tube and consequently not reach the mass spectrum detector.

Figure R21. TG-MS data of 2D PhenPtCl₂ crystal for analysis of mass loss in different stages.

Subsequent to the 500 °C mark, we did not observe significant weight loss stages. Instead, a gradual mass loss was identified, associated with the decomposition of amorphous carbon materials. This was substantiated by the presence of clear mass spectrum peaks at 12 and 44 (CO₂). These observations indicate that structural variations occurring at different annealing temperatures are likely to influence the HER performance, emphasizing the importance of optimizing the HER performance of the 2D PhenPtCl₂ material.

All these data have been added into Figs. 2a on Page 10 of the revised manuscript file, and the related discussion has been added on Page 8-9 of the revised manuscript file.

4. Following Q3, the authors, in fact, prepared several different materials by annealing ‘PhenPtCl₂’ at different temperatures. These materials show huge structural and compositional differences. Thus, whether these materials can be compared to explain the HER performance difference is questionable.

Response:

In response to comment #3, we conducted a thorough analysis of the weight loss at different stages using TG-MS to unravel the structural evolution of 2D PhenPtCl₂

crystals at varying temperature stages. Our investigations revealed a significant shift in the crystal structure after 200 °C, resulting in the absence of freely dynamic organic coordination compounds capable of leaving and coordinating Cl⁻. Consequently, the material transitioned into an amorphous compound with a 4-coordination structure, leading to the inactivation of active sites and deterioration in electron transport. This structural change ultimately led to a gradual decline in the HER performance.

Subsequently, we systematically explored the HER performances of 2D PhenPtCl₂ crystals at different temperature stages. Our investigations confirmed that the structural evolution of 2D PhenPtCl₂ crystals directly influences their HER performances. Particularly, after 200 °C, there was a sharp deterioration in HER performance, indicating the loss of vitality in active site regulation. The dynamic coordination capability ceased to function, leading to the loss of the material's excellent HER activity. This pivotal insight laid the foundation for the subsequent analysis of the outstanding HER performance exhibited by 2D PhenPtCl₂ samples at 100 °C.

Through in-situ Raman and in-situ XPS analyses, we substantiated that 2D PhenPtCl₂ samples annealed at 100 °C are undergoing a dynamic coordination process within the electrolyte. The removal and retention of Cl⁻ ions within the 2D PhenPtCl₂ crystal induced dynamic changes in Pt coordination structures. Notably, unsaturated 2-coordinated Pt emerged as a critical factor significantly enhancing the HER performance, underscoring the intricate interplay between structural dynamics and catalytic activity.

5. Following Q3, as shown in TG and Raman results (Fig. 2c, S12), the materials at 25 and 100 °C should be the same (Line 168), but why do they show such a huge difference in HER performance? The material prepared by annealing 'PhenPtCl₂' at 100 °C deserves further study.

Response:

Thanks for the comments. We here make the clarification of the 2D PhenPtCl₂ samples obtained through annealing at 100 °C for more details. It is important to note that the best HER performance of these samples, as highlighted by the reviewer, may

indeed be attributed to factors such as improved crystal structure and a high purity following annealing. Moreover, the references to TGA data and Raman spectra further support the notion that there is no structural decomposition or change before 200 °C, reinforcing the consistency between samples annealed at 25 °C and 100 °C.

To provide a comprehensive understanding of the structural equivalency of these samples, we have conducted additional XRD and XPS experiments. As depicted in Figure R22, the XRD data confirm the presence of identical crystal diffraction peaks in the 2D PhenPtCl₂ samples annealed at 25 °C and 100 °C, respectively. Additionally, the XPS analysis presented in Figure R23 reveals no discernible differences in valence states and peak positions between these two sets of samples.

In light of these observations and analyses, it is reasonable to conclude that the 2D PhenPtCl₂ samples annealed at 25 °C and 100 °C represent the same crystalline structure, with no evidence of crystal structure evolution occurring at 100 °C.

Figure R22. XRD data of 2D PhenPtCl₂ samples at 25 °C and 100 °C.

Figure R23. XPS data of 2D PhenPtCl₂ samples at 25 °C and 100 °C in (a) C 1s, (b) Cl 2p, (c) N 1s and (d) Pt 4f.

It is crucial to emphasize that XAFS and XPS are specifically derived from 2D PhenPtCl₂ samples subjected to annealing at 100 °C. To provide an exhaustive characterization of these 2D PhenPtCl₂ samples at 100 °C, we have introduced additional data and images as outlined below.

New XPS data, as illustrated in Figure R24, offer an in-depth exploration of the composition and valence states of the 2D PhenPtCl₂ samples after annealing at 100 °C. Furthermore, Figures R25-27 feature images and data encompassing iDPC-STEM, EDS elemental maps, XRD, and SEM. These supplementary figures provide a comprehensive view of the structural and elemental characteristics of the 2D PhenPtCl₂ samples following annealing at 100 °C.

Importantly, our analysis has revealed that there are no discernible distinctions between the 2D PhenPtCl₂ samples annealed at 100 °C and those annealed at 25 °C. This consistency underscores that the improvements in HER performance are not the result of structural variances but may be attributed to factors such as enhanced crystal quality and increased purity, as previously discussed.

All these data have been added into Figs. 2c-2f on Page 10 of the revised manuscript file and Supplementary Fig. 3 on Page 7 and Supplementary Fig. 14 on Page 13 of the revised SI file, and the related discussion has been added on Page 9 and 11 of the revised manuscript file.

Figure R24. XPS data of 2D PhenPtCl₂ samples at 100 °C in (a) C 1s, (b) Cl 2p, (c) N 1s and (d) Pt 4f.

Figure R25. (a) iDPC-STEM images and (b) EDS mapping of 2D PhenPtCl₂ samples at 100 °C.

Figure R26. XRD data of 2D PhenPtCl₂ samples at 100 °C.

Figure R27. SEM images of 2D PhenPtCl₂ samples at 100 °C.

6. The authors claimed a dynamic evolution of Pt-Cl during HER stability tests based on the Raman results (Fig. 4a), but why do other Raman peaks, such as Pt-N, decrease at the same time? If the original material decomposed (Line 266), what's the exact final material? Since the ΔG_{H}^* of Phen-Pt with different Cl coordination varies greatly (Fig. 4e, 4f), why does the HER remain stable during the evolution of Pt-Cl? Does the material undergo the same dynamic evolution of Pt-Cl in electrolytes with different acidity?

Response:

In response to the reviewer's query, we conducted a thorough analysis of the Raman vibration peak intensity associated with the Pt-Cl bond (I_{Pt-Cl}) and the Pt-N bond (I_{Pt-N}), as illustrated in Figure R28. To our surprise, we observed a gradual reduction in the ratio of I_{Pt-Cl}/I_{Pt-N} during continuous electrocatalysis. This notable observation strongly supports the dissociation of Cl⁻ during the catalytic process and provides a clear explanation for the dynamic coordination of Cl⁻ throughout the catalytic reaction. Based on the decreased I_{Pt-N} , it is possible that this reduction can be attributed to deterioration on the integrity of the crystal structure during the electrolysis process. Nevertheless, the original crystal structure is maintained.

Figure R28. Raman intensity ratio of $I_{\text{Pt-Cl}}/I_{\text{Pt-N}}$ for 2D PhenPtCl₂ samples in the catalytic process.

Figure R29. iDPC-STEM images of 2D PhenPtCl₂ samples at 100 °C after the stability test in 2 h.

To further confirm the atomic structure of the 2D PhenPtCl₂ samples at 100°C after the electrocatalytic reaction, we carried out the iDPC-STEM imaging, as presented in Figure R29. As shown in the atomic-scale image, although the crystal structure has partially decomposed, it still retains the original crystal structure, characterized by the

presence of Phen-Pt-Cl₂. Notably, the Raman vibration peaks associated with the Pt-Cl bond and Pt-N bond, as well as the presence of Pt-N and Pt-Cl, remain intact. Therefore, it can be concluded that upon the completion of the reaction and re-equilibrium with Cl⁻ in the electrolyte, the crystal structure reverts to PhenPtCl₂.

The calculated ΔG_{H}^* values for Phen-Pt with different Cl coordinations have revealed significant variations. In the context of the HER tests, it is evident that the coordination of Cl⁻ profoundly influences the electrocatalytic performance of 2D PhenPtCl₂ samples annealed at 100 °C. To elucidate this effect, we conducted polarization curve tests on these samples with varying scan times, as depicted in Figure R30.

Upon the initial scan, the 2D PhenPtCl₂ samples at 100 °C exhibited relatively weak catalytic activity. However, during the second scan, they demonstrated a notable improvement in catalytic HER performance. As the number of scan times increased, the samples underwent a process of adaptation, and after approximately five scans, they reached a state of equilibrium, consistently showcasing excellent HER performance. This observation underscores the dynamic and adaptive nature of the catalytic behavior of 2D PhenPtCl₂ samples at 100 °C, reinforcing the influence of Cl⁻ coordination on their electrocatalytic performances.

Figure R30. Cathodic polarization curves of 2D PhenPtCl₂ nanosheets with different

times.

To gain deeper insights into the dynamic interaction between Pt, primarily coordinated with Phen ligands, and Cl^- ions in the surrounding electrolyte, especially Cl^- ions, we employed in-situ XPS to analyze the evolution of Pt valence and changes in Cl content throughout the entire electrocatalytic process, as presented in Figure R31. Analysis of the XPS peaks of Pt 4f revealed that the valence of Pt transitioned from an initial valence of +4 to +2 during the electrocatalytic HER process in the in-situ XPS test. Following the conclusion of the HER, Pt retained +3 valence state. This dynamic evolution of Pt valence states, with a notable two-coordinate Pt atom at +2 valence, underscores the influence of Pt coordination on the excellent HER performances.

Simultaneously, the content of Cl, determined by the peak intensity of Cl 2p in the 2D PhenPtCl₂ samples, displayed an intriguing pattern. Initially, it decreased and then increased over the course of the entire electrocatalytic process. Notably, while the valence state of Pt and the peak intensity of Cl 2p did not fully revert to their original states after the HER, this is likely due to the competitive processes involving other cations (H^+) in the electrolyte, which may hinder Pt from continued coordination with Cl^- . Alternatively, this variation could be attributed to potential damage to certain crystal structures.

Of particular significance, the intensity of the Cl 2p peak at 197.7 eV and the binding energy of Pt 4f_{7/2} for the 2D PhenPtCl₂ samples at 100 °C throughout the electrocatalytic process, and 2 hours after the HER in the acidic electrolyte, revealed distinct trends. The Cl 2p peak intensity experienced an initial rapid decrease, followed by partial recovery after the HER concluded. Concurrently, the binding energy of Pt 4f exhibited similar fluctuations, initially decreasing and subsequently rising (Figure R31c). These findings are instrumental in elucidating that Cl^- ions are removed and retained during the electrocatalysis process, with subsequent re-coordination with Pt during post-HER. These results unequivocally confirm that the removal and retention of Cl^- in 2D PhenPtCl₂ samples trigger dynamic changes in the Pt coordination structure. The Pt active sites exhibit dynamic coordination during the electrocatalytic process, with unsaturated, two-coordinated Pt contributing significantly to enhanced HER

performance.

Figure R31. In-situ XPS at (a) Pt 4f peak and (b) Cl 2p peak of 2D PhenPtCl₂ samples at 100 °C during the whole catalytic process and 2 h after HER in the acidic electrolyte. (c) Intensity of Cl 2p peak at 197.7 eV and binding energy of Pt 4f_{7/2} for 2D PhenPtCl₂ samples at 100 °C during the whole catalytic process and 2 h after HER in the acidic electrolyte.

Additionally, we conducted in-situ XPS tests to evaluate the valence state of the Pt element and the Cl content in neutral and alkaline electrolytes during the electrocatalytic HER, as shown in Figure R32. The observed changes in Cl content and Pt valence state are consistent with the patterns observed in acidic electrolytes. Although the shift in Cl content is less pronounced in neutral and alkaline electrolytes,

this difference may be attributed to the higher likelihood of H^+ dissociating Cl^- in acidic solutions. These findings collectively emphasize that during the electrocatalytic HER, the active Pt site at the core of 2D PhenPtCl₂ crystals at 100 °C is engaged in a dynamic coordination equilibrium process. The intermediate state, Phen-Pt with a +2 valence, plays a pivotal role in promoting efficient HER.

All these data have been added into Figs. 4b-4e on Page 17 of the revised manuscript file and Supplementary Fig. 23 and Fig. 25 on Page 18-19 of the revised SI file, and the related discussion has been added on Page 15-16 of the revised manuscript file.

Figure R32. In-situ XPS data at (a) Pt 4f peak and (b) Cl 2p peak of 2D PhenPtCl₂ samples at 100 °C during the whole catalytic process and 2 h after the catalytic reaction in the neutral electrolyte. In-situ XPS data at (c) Pt 4f peak and (d) Cl 2p peak of 2D PhenPtCl₂ samples at 100 °C during the whole catalytic process and 2 h after the catalytic reaction in the alkaline electrolyte.

REVIEWERS' COMMENTS

Reviewer #1 (Remarks to the Author):

The authors have addressed the reviewer's questions and the manuscript can be accepted as it is.

Reviewer #2 (Remarks to the Author):

I have carefully reviewed the revised version of the manuscript and Point-by-point Response to the Comments. Basically met the revision requirements and agreed to be published.

Reviewer #3 (Remarks to the Author):

The manuscript is ready for publication